# Photoperiodic control of the *Arabidopsis* proteome reveals a translational coincidence mechanism

Daniel D Seaton[1,†,‡] (iD), Alexander Graf[2,†,§] (iD), Katja Baerenfaller[2,¶] (iD), Mark Stitt[3] (iD), Andrew J Millar[1,*] (iD) & Wilhelm Gruissem[2,**] (iD)

## Abstract

**Plants respond to seasonal cues such as the photoperiod, to adapt to current conditions and to prepare for environmental changes in the season to come. To assess photoperiodic responses at the protein level, we quantified the proteome of the model plant *Arabidopsis thaliana* by mass spectrometry across four photoperiods. This revealed coordinated changes of abundance in proteins of photosynthesis, primary and secondary metabolism, including pigment biosynthesis, consistent with higher metabolic activity in long photoperiods. Higher translation rates in the day than the night likely contribute to these changes, via an interaction with rhythmic changes in RNA abundance. Photoperiodic control of protein levels might be greatest only if high translation rates coincide with high transcript levels in some photoperiods. We term this proposed mechanism "translational coincidence", mathematically model its components, and demonstrate its effect on the *Arabidopsis* proteome. Datasets from a green alga and a cyanobacterium suggest that translational coincidence contributes to seasonal control of the proteome in many phototrophic organisms. This may explain why many transcripts but not their cognate proteins exhibit diurnal rhythms.**

**Keywords** circadian rhythms; metabolism; photoperiod; proteomics; seasonality

**Subject Categories** Plant Biology; Post-translational Modifications, Proteolysis & Proteomics; Quantitative Biology & Dynamical Systems

**Mol Syst Biol. (2018) 14: e7962**

## Introduction

Changes in photoperiod have wide-ranging effects on the physiology, metabolism and development of many species, from migration and hibernation in birds and mammals to diapause in insects (Saunders, 2013; Dardente *et al*, 2014). In *Arabidopsis*, these responses include changes in flowering time (Yanovsky & Kay, 2002; Salazar *et al*, 2009), hypocotyl elongation (Nozue *et al*, 2007), freezing tolerance (Lee & Thomashow, 2012) stomatal opening (Kinoshita *et al*, 2011), C-allocation and growth (Sulpice *et al*, 2014; Mengin *et al*, 2017). These diverse responses to photoperiod allow plants to adjust to the predictable environmental changes that accompany the changing seasons. Here, we investigate photoperiod responses at the proteome level and ask two related questions: How does the proteome change with photoperiod, and which regulatory mechanisms contribute to changes in protein abundance across photoperiods?

Plants use daytime sunlight as a source of energy to drive photosynthesis. As a result, day length has strong effects on metabolism and growth, with increasing photoperiod length leading to a progressive increase in the rate of growth, which is often accompanied by increased levels of many metabolites (Gibon *et al*, 2004; Sulpice *et al*, 2014). Furthermore, growth under different photoperiods places different demands on plant physiology and metabolism. In *Arabidopsis*, for example, the major carbon source at night comes from the mobilisation of transient starch that is accumulated in leaf cell chloroplasts during the light period (Smith & Stitt, 2007; Graf & Smith, 2011; Stitt & Zeeman, 2012). Rates of starch mobilisation to sucrose are higher during short nights relative to long nights, whereas daytime partitioning of photosynthate into starch is higher during short compared to long days, and these rates change progressively with photoperiod duration (Sulpice *et al*, 2014; Mengin *et al*, 2017). Pathways of primary carbon metabolism might be expected to change in concert with the availability of carbon and its partitioning. Secondary metabolism will be affected not only by changing availability of primary carbon substrates, but the accumulation of

1 SynthSys and School of Biological Sciences, University of Edinburgh, Edinburgh, UK
2 Department of Biology, Institute of Molecular Plant Biology, ETH Zurich, Zurich, Switzerland
3 System Regulation Group, Max Planck Institute of Molecular Plant Physiology, Potsdam-Golm, Germany
*Corresponding author. Tel: +44 131 651 3325; Email: andrew.millar@ed.ac.uk
**Corresponding author. Tel: +41 44 632 0857; Email: wgruisse@ethz.ch
†These authors contributed equally to this work
‡Present address: European Molecular Biology Laboratory, European Bioinformatics Institute, Hinxton, UK
§Present address: Plant Proteomics Group, Max Planck Institute of Molecular Plant Physiology, Potsdam-Golm, Germany
¶Present address: Swiss Institute of Allergy and Asthma Research, University of Zürich, Davos, Switzerland

certain secondary metabolites will also be affected by seasonal selective pressures, for example for compounds that defend against seasonal pests and pathogens (Textor & Gershenzon, 2009). In general, it is not well understood how investment in protein synthesis balances these different demands.

In previous studies, we analysed starch turnover, metabolite levels and the rates and diurnal distribution of growth (Sulpice *et al*, 2014), the transcriptional response of central clock genes and the dawn transcriptome (Flis *et al*, 2016) in *Arabidopsis* Col-0 growing in a 6-, 8-, 12- or 18-h photoperiod. Quantitative proteomics can characterise changes in protein abundance with photoperiod, as was recently reported for a small number of *Arabidopsis* proteins (Baerenfaller *et al*, 2015). Here, we measured the regulation of the *Arabidopsis* proteome using quantitative mass spectrometry and identified >1,700 proteins that change in abundance across four photoperiods. The changes revealed adjustments to growth in different photoperiods, with coordinated changes of protein investment in photosynthesis and primary carbon metabolism, consistent with the higher demand placed on these pathways under long photoperiods.

The mechanisms underlying photoperiod-responsive, physiological changes involve the integration of diel (daily) signals from the environment with timing information from the circadian clock. Several response mechanisms, including flowering time and elongation growth, share a common form known as the "external coincidence" mechanism and have been sufficiently characterised to inform quantitative, mathematical models (Keily *et al*, 2013; Seaton *et al*, 2015). Briefly, they involve intermediate transcriptional regulators, such as *CONSTANS*, *FKF1* and *CDF1*, which are among the > 30% of clock-controlled transcripts in *Arabidopsis* (Edwards, 2006; Covington *et al*, 2008; Michael *et al*, 2008b). Environmental signals such as light or darkness alter the stability or activity of their cognate proteins if the timing of these changes coincides with the phase of rhythmic expression, in which case a photoperiodic response is observed (Yanovsky & Kay, 2002; Salazar *et al*, 2009). These transcriptional cascades are well-studied, specific examples of photoperiod signalling. However, it is unclear whether these or equivalent coincidence mechanisms act in a general way to mediate the many photoperiodic responses observed in plant physiology. Their canonical phenotypes, especially seasonal reproduction, are the most important, known effects of plant circadian regulation. This suggests a potential role for photoperiod responses as a driving force for the evolution of pervasive circadian regulation across the genome (Millar, 2016).

Photoperiodic regulation of the proteome could be driven by changes in RNA levels, translation and/or protein turnover. We recently showed that there are major photoperiod-dependent changes in global transcript abundance that affect large sets of genes involved in metabolism and growth (Flis *et al*, 2016), and that transcripts with different levels in long and short photoperiods are overrepresented in categories such as flavonoid biosynthesis and sugar transport (Baerenfaller *et al*, 2015). In the absence of compensating regulation, these changes in RNA abundance are expected to result in changes in protein level. At the post-transcriptional level, multiple lines of evidence have demonstrated changes in the rate of plant protein synthesis in response to light, with translation proceeding more rapidly during the day than during the night (Piques *et al*, 2009; Juntawong & Bailey-Serres, 2012; Liu *et al*, 2012; Pal *et al*, 2013; Ishihara *et al*, 2015; Missra *et al*, 2015). This translational regulation suggests that the profile of protein synthesis across the diel cycle will depend on the duration of the light period, even without circadian regulation. However, higher rates of translation in the light on their own would tend to lead to a general increase in the abundance of proteins. The question arises whether the light-dependent increase in translation might interact with the widespread rhythmicity in RNA levels, which affects up to 50% of genes in *Arabidopsis* (Bläsing *et al*, 2005; Michael *et al*, 2008b; Baerenfaller *et al*, 2012). These known diel changes of translation and transcript levels prompted a simple, data-driven model that predicts how these two well-characterised effects might systematically alter protein levels. Briefly, our model suggests that transcripts that peak early in the 24-h cycle will be efficiently translated in long and short photoperiods, whereas transcripts that peak later in the 24-h cycle will be efficiently translated in long but not in short photoperiods. The proposed mechanism, which we termed "translational coincidence", was tested using our quantitative data on protein abundance across a range of photoperiods.

Our data implicate multiple mechanisms in the regulation of protein abundance with photoperiod. Changes in RNA abundance contribute to some changes in protein abundance. However, our data are also consistent with the predicted effects of translational coincidence affecting hundreds of proteins in *Arabidopsis*. Analysis of existing experimental data from cyanobacteria and algae indicate that translational coincidence most likely applies broadly, across many phototrophic organisms. These results reveal new insights into photoperiod responses in plants, and the mechanisms that drive them.

# Results

### Photoperiod length affect protein abundance

The effect of changes in photoperiod length on the proteome of *Arabidopsis* was analysed in 30-day-old wild-type plants grown for 9 days in four different light/dark cycles equalling 6-h, 8-h, 12-h and 18-h photoperiods. Full rosettes were harvested at the end of the day (ED) (Fig 1A; see Materials and Methods). Protein abundance at ED most directly captures the impact of light period duration on the proteome. Previous studies found only few proteins that significantly changed in abundance between end of night (EN) and ED in a 16-h or 8-h photoperiod (Baerenfaller *et al*, 2012, 2015), as expected if most of the detected proteins have long half-lives [median in one recent study was > 6 days (Li *et al*, 2017)]. It is therefore likely that for most proteins, their abundance at ED reflects their abundance over the entire 24-h cycle.

Quantitative data was obtained for 4,344 proteins (Table EV1), which increased the coverage of enzymes in all metabolic pathways (Table EV2) compared to previous reports of *Arabidopsis* leaf 6 proteins quantified at four leaf growth stages and in three different growth conditions (Baerenfaller *et al*, 2012, 2015). Proteomic studies, especially with plants, tend to show overrepresentation of abundant proteins in the set of quantified proteins. While this is also true for the present data set, low abundant proteins annotated in the KEGG pathways of basal transcription factors (5) and hormone signalling (13) were also quantified (Table EV2).

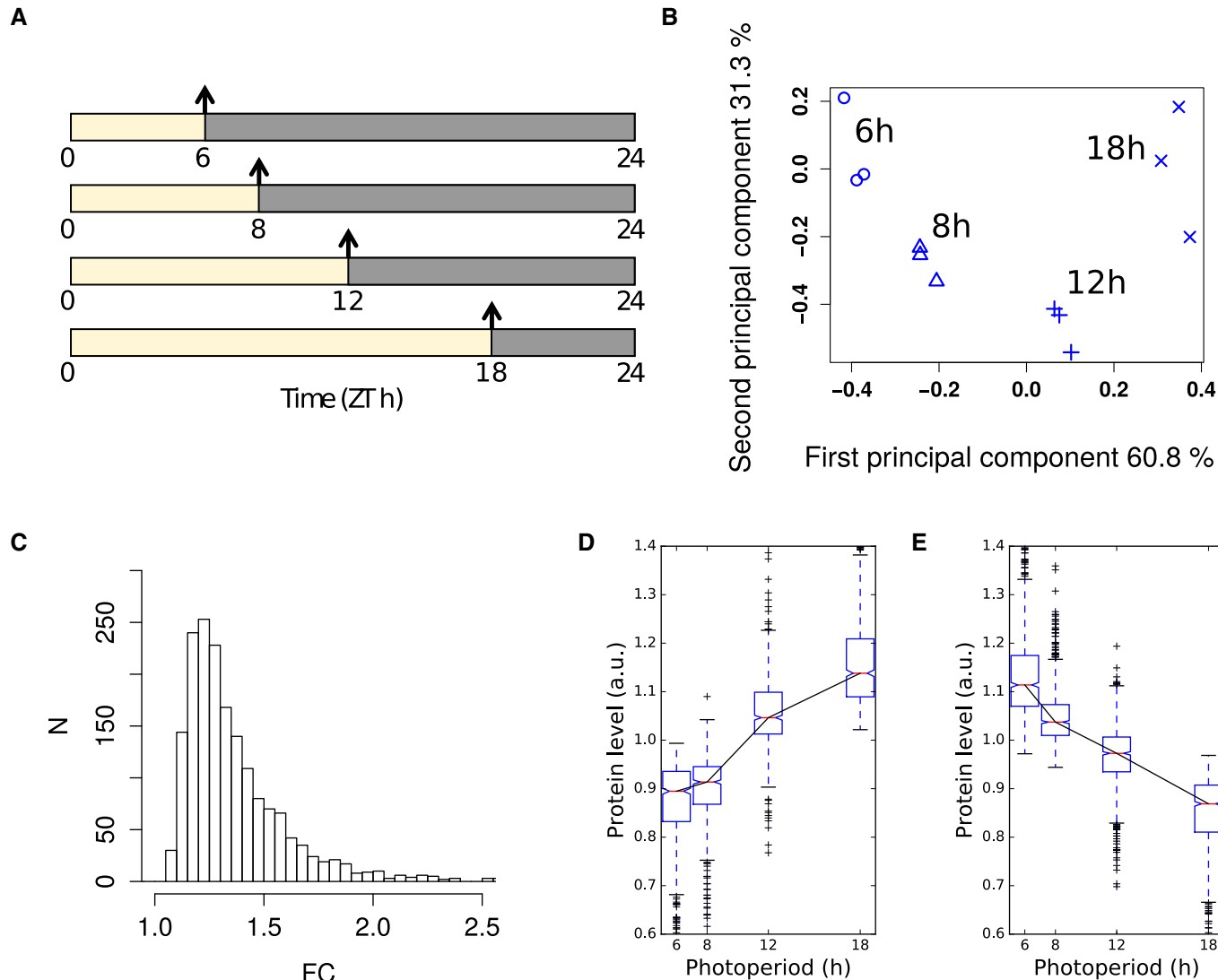

**Figure 1. Overview of photoperiod proteome dataset.**

A   Summary of sampling protocol. Samples were taken at the end of the day (arrows) from 30-day-old plants grown for 9 days in photoperiods of 6-, 8-, 12- and 18-h duration.

B   Principal component analysis of proteomics dataset, showing % variance explained by each component. The three biological replicates from each photoperiod cluster together.

C   Histogram of maximal fold changes (FC) across proteins identified as significantly changing with photoperiod ($P < 0.05$).

D   Progressive changes in abundance across photoperiods for proteins exhibiting significant changes with photoperiod. Protein abundance across photoperiods for proteins which increase in abundance in longer photoperiods. Protein abundance for each protein was mean-normalised.

E   As in (D), for proteins which decrease in abundance in longer photoperiods.

Data information: Boxes span the interquartile range. Whiskers span 1.5 times the interquartile range.

The variation between biological samples was comparable in all four photoperiods. Moreover, the average coefficient of variation (between 0.059 and 0.074) is very low for a proteomics data set. Principal component analysis completely separated the samples according to the photoperiods while biological replicates remained grouped together, confirming the reproducibility of the dataset (Fig 1B). The first and second principal components together accounted for 95.7% of the total variation.

We found 1,781 proteins (41%) that changed significantly ($P < 0.05$, ANOVA) in abundance between the four photoperiods

(Table EV3). Of these, 389 proteins had a maximum fold change (FC) greater than 1.5 (Fig 1C). A comparison between four comparable growth stages of *Arabidopsis* leaf 6 in plants grown in a 8-h or 16-h photoperiod also showed that 192 of 1,200 quantified proteins had significant abundance changes ($P < 0.05$), with maximum fold changes of at least 1.5 for 83 proteins (Baerenfaller *et al*, 2015). The larger number of changing proteins we identified could be explained by the larger span of photoperiods as well as differences in growth regimes and sampled tissues. However, the fold changes of the proteins with significant changes at a *P*-value threshold of 0.05 in

both datasets were positively correlated ($\rho = 0.46$), indicating similar trends. Here, 757 proteins had higher abundance in longer photoperiods while 1,024 proteins showed lower abundance. Boxplots of all photoperiod-responsive proteins revealed that proteins were up-regulated mainly between the three longest photoperiods (8–18 h) while only few proteins increased in abundance between the two shortest photoperiods (6–8 h; Fig 1D). In contrast, the decrease in protein abundance was more evenly distributed across all photoperiods (Fig 1E). The progressive change of protein abundance is also reflected by a pairwise comparison between photoperiods. Only few proteins change significantly when comparing 8 h vs. 6 h (12), 12 h vs. 8 h (184) or 18 h vs. 12 h (177) (Tables EV3 and EV4). However, high numbers are observed when comparing 18 h vs. 8 h (1,035) or 18 h vs. 6 h (1,452). This resembles the progressive change in transcript levels between a 6-h, 8-h, 12-h and 18-h photoperiod (Flis *et al*, 2016).

More than half (50.3%) of the observed changes in protein abundance was below a FC of 1.3, with a mean FC of approximately 1.2 (Fig 1C, Table EV4). While these changes are relatively small, their potential biological significance is illustrated by the enrichment of gene ontology (GO) terms within narrow ranges of FC. This was assessed by binning proteins into FC windows of 0.2. Each bin was analysed for overrepresentation of GO terms compared to all quantified proteins. The overrepresentation analysis revealed that enriched GO annotations can be found in each of the applied FC bins (Fig 2A, Table EV5), and in most cases, a particular GO category was overrepresented in a specific FC bin. For example, 338 proteins annotated to the GO category translation were found in the whole data set. In a narrow bin ranging from 1.1 to 1.3 FC, 106 of these proteins were identified as down-regulated in longer photoperiods. This results in a significant overrepresentation of translation-related proteins in this FC bin (Fisher's exact test, $P < 10^{-18}$). High enrichments in specific FC bins were also observed for other GO categories including the tricarboxylic acid (TCA) cycle (GO:0006099, bin up 1.1–1.3), translational elongation (GO:0006414, bin down 1.1–1.3), ribosome biogenesis (GO:0042254, bin down 1.0–1.2), glucosinolate biosynthesis (GO:0019761, bin up 1.2–1.4) and indoleacetic acid biosynthesis (GO:0009684, bin up 1.3–1.5). Heatmaps of the overrepresented GO terms in the different FC bins further illustrate that changes in abundance of functionally related proteins are highly orchestrated in a narrow FC window (Fig 2B and C).

## Photoperiod length affects photosynthesis, metabolism and growth

As photoperiods become longer, plant metabolism and energy management are adjusted to the increased availability of light and shorter heterotrophic intervals during the night (Sulpice *et al*, 2014; Baerenfaller *et al*, 2015). Changes in the plant proteome reflect this plasticity at multiple levels, from primary photosynthesis to secondary metabolism, cellular regulation and growth. For example, we quantified 57 of 77 proteins annotated in the KEGG pathway (Kanehisa *et al*, 2016) of photosynthesis (ath00195) and 22 of the quantified proteins were more abundant in longer photoperiods (Table EV6; Appendix Fig S1). These changes affect all complexes of the electron transport chain, several subunits of the ATP synthase complex as well as ferredoxin 1 (FD1) and ferredoxin-NADP-oxido-reductase 1 (FNR1; Fig 3A and B). While most proteins in our

dataset showed a gradual change in abundance over all photoperiods (Fig 1D and E), changes in abundance of photosystem I and II related proteins occurred predominantly between the 6-h and 12-h photoperiods, beyond which the protein levels reached a plateau (Fig 3A and B). The only proteins of the photosynthetic electron transport chain with lower abundance in long photoperiods are plastocyanin 1 and 2 (PETE1 and PETE2), which are responsible for transporting electrons from the cytochrome-b$_6$f-complex to photosystem (PS) I. Similar concerted changes in protein abundance were also observed in the light harvesting and chlorophyll binding complexes (LHCII) surrounding PSII (Fig 3C; Tables EV3 and EV6), which are correlated with changes in their transcript levels (Baerenfaller *et al*, 2015; Flis *et al*, 2016).

Differential changes in enzyme abundance were found for isoprenoid metabolic pathways, including biosynthesis of chlorophyll. For example, 13 enzymes involved in chlorophyll biosynthesis were down-regulated in longer photoperiods (Table EV6, Appendix Fig S2). These included enzymes in heme biosynthesis (HEMA1, HEMB1, HEME2 and HEMG2) as well as the magnesium chelatase GUN5 and the NADPH:protochlorophyllide oxidoreductases PORB and PORC (Fig 3D). In contrast, increased enzyme abundance was observed for the red chlorophyll catabolite reductase ACD2, which catalyses a key reaction of chlorophyll catabolism.

Enzymes in primary carbon metabolism were broadly up-regulated in longer photoperiods. Proteins with higher abundance in longer photoperiods are enriched for the KEGG pathways of carbon fixation (ath00710), the TCA cycle (ath00020) and starch and sucrose metabolism (ath00500; Table EV6; Appendix Figs S3–S5). Their abundance changes are highly orchestrated, and this was especially pronounced for the TCA and Calvin–Benson cycles (Table EV6). Similarly, proteins in sucrose metabolism including sucrose synthesis, transport and degradation accumulated to higher levels in longer photoperiods (Fig 4A; Appendix Fig S5). Protein abundance in the metabolic pathways of starch synthesis and degradation was also strongly affected by photoperiod length. For example, proteins such as APL3, one of the two regulatory subunits of plastid ADP-glucose pyrophosphorylase (AGPase) that catalyses the first committed step in starch synthesis, and plastid phosphoglucomutase (PGM1) that regulates the partitioning of carbon into starch (Fernie *et al*, 2001), are strongly increased while APL1 is decreased in the longest photoperiod (Fig 4A). Several key enzymes for starch degradation also accumulated to higher levels with increasing photoperiod length (Fig 4B), consistent with a faster rate of starch degradation during the night in long photoperiods (Smith & Stitt, 2007; Sulpice *et al*, 2014; Baerenfaller *et al*, 2015).

Our data show that *Arabidopsis* can also reprogram sulphur metabolism (Fig 4C) and adjust the abundance of enzymes for lipid metabolism to the prevailing photoperiod length (Table EV6, Appendix Figs S6 and S7). The changes in abundance of sulphate assimilating enzymes indicate a shift from the synthesis of primary to secondary sulphur-containing metabolites in longer photoperiods, including a concerted increase in abundance of enzymes involved in glucosinolate biosynthesis (Fig 4C, Table EV6; Appendix Fig S9). This is consistent with increased availability of resources for the production of defence-related metabolites in plants growing in long photoperiods (del Carmen Martínez-Ballesta *et al*, 2013; Baerenfaller *et al*, 2015). Several *Arabidopsis* enzymes in fatty acid degradation are more abundant in long photoperiods (Table EV6,

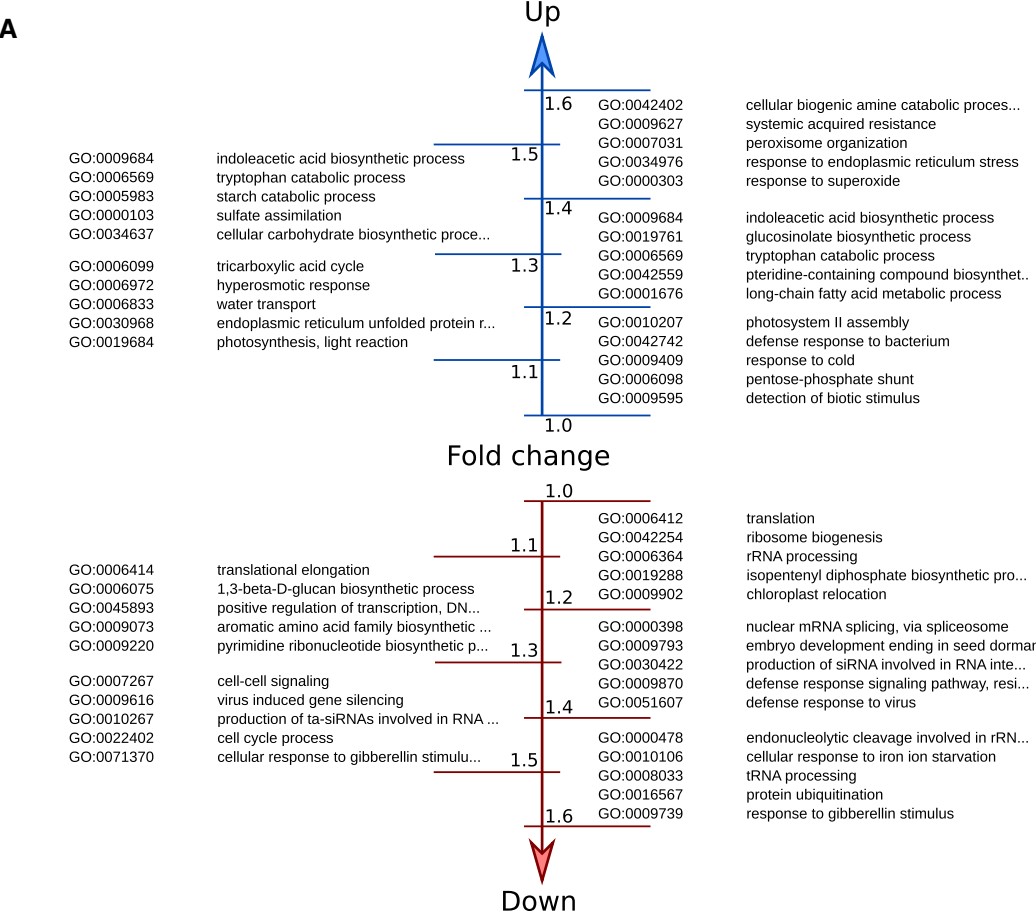

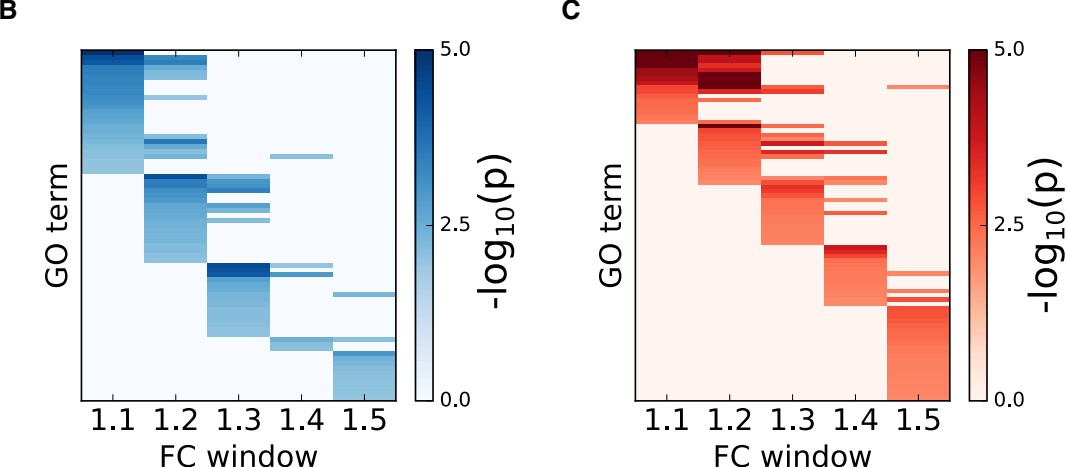

**Figure 2.  Enrichment of GO terms in fold change (FC) windows for proteins up- and down-regulated with increasing photoperiod.**

A   Five high-scoring GO enrichments of proteins are listed for each FC window.
B   Heatmap of GO enrichments for each FC window for significantly up-regulated proteins (enrichment scored by −log10(*P*-value) of Fisher's exact test). Only GO enrichments in FC windows up to 1.5 are shown. Complete enrichments across all FC windows are provided in Table EV5.
C   As in (B), for significantly down-regulated proteins.

Appendix Fig S6) while levels of several enzymes for fatty acid synthesis are reduced (Table EV6, Appendix Fig S7). This indicates that in longer photoperiods *Arabidopsis* has a higher capacity for beta-oxidation of fatty acids, consistent with the turnover of approximately 4% of the total fatty acids in one diel cycle (Bao *et al*, 2000).

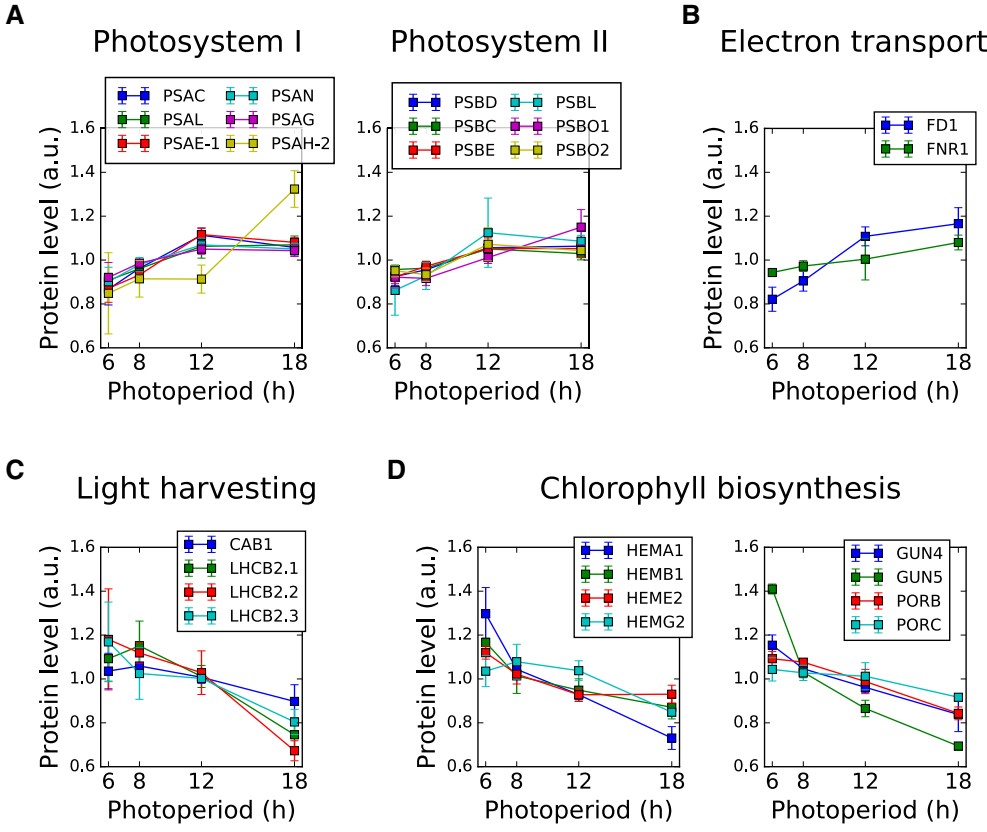

**Figure 3.  Photoperiod modulates protein levels in processes and complexes involved in photosynthesis.**

A   Significantly up-regulated proteins in photosystem I and II: photosystem I subunits C, L, E, N, G, H (PSAC, PSAL, PSAE-1, PSAN, PSAG, PSAH-2), photosystem II subunits D, C, E, L, O1, O2 (PSBD, PSBC, PSBE, PSBL, PSBO1, PSBO2).

B   Significantly up-regulated proteins in the electron transport chain: ferredoxin 1 (FD1) and ferredoxin-NADP-oxidoreductase 1 (FNR1).

C   Significantly down-regulated proteins in the light harvesting complex: Chlorophyll a-b binding proteins 1, 2.1, 2.2 and 2.3 (CAB1, LHCB2.1, LHCB2.2, LHCB2.30).

D   Significantly down-regulated proteins in chlorophyll biosynthesis, including enzymes involved in heme biosynthesis (glutamyl-tRNA reductase 1, HEMA1; delta-aminolevulinic acid dehydratase 1, HEMB1; uroporphyrinogen decarboxylase 2, HEME2), as well as tetrapyrrole-binding protein (GUN4), magnesium chelatase (GUN5) and NADPH:protochlorophyllide oxidoreductases (PORB, PORC).

Data information: Error bars represent the standard deviation ($n = 3$).

Increased photoperiod length results in a highly active metabolic state of the *Arabidopsis* rosette leaves (Sulpice *et al*, 2014). Our results show that this was correlated with the down-regulation of pathways related to cell cycle and protein biosynthesis. A GO term overrepresentation analysis using photoperiod-responsive proteins with lower abundance in long photoperiods revealed that most of the 41 significantly enriched GO categories are related to transcription, translation and cell cycle (Table EV7). The concerted changes in protein abundance of the translation machinery were particularly striking. Among the 33 quantified proteins annotated for ribosome biogenesis, 19 were less abundant in longer photoperiods(Table EV6 and Fig EV1A) and no proteins in this category had increased levels. We also quantified 151 proteins annotated in the KEGG pathway for ribosomes, of which 85 were less abundant in longer photoperiods (Table EV6 and Fig EV1B). Only two ribosomal proteins, RPS6A and RPS6B that are functionally redundant and essential for the 40S ribosomal subunit (Creff *et al*, 2010), are more abundant in longer photoperiods.

Together, these results are consistent with the reduced vegetative growth period and early flowering of *Arabidopsis* plants in long photoperiods, which is compensated by high metabolic activity of the smaller rosette (see Appendix Supplemental Text for an extended description of additional functional categories displaying significant changes).

### Correlated changes in transcript and protein abundance

Transcriptional regulation is one potential mechanism for explaining changes in protein levels across photoperiods. The *Arabidopsis* transcriptome at EN and ED shows large photoperiod-dependent changes (Flis *et al*, 2016). We compared photoperiod-dependent changes in protein abundance at ED to photoperiod-dependent transcriptome changes at ED and EN. Only transcript-protein pairs were considered that showed significant changes in both transcript ($P < 0.05$, FC > 1.5) and protein level ($P < 0.05$). Using these conditions, 421 and 390 transcript:protein pairs were selected at ED and EN, respectively. The protein abundance changes at ED

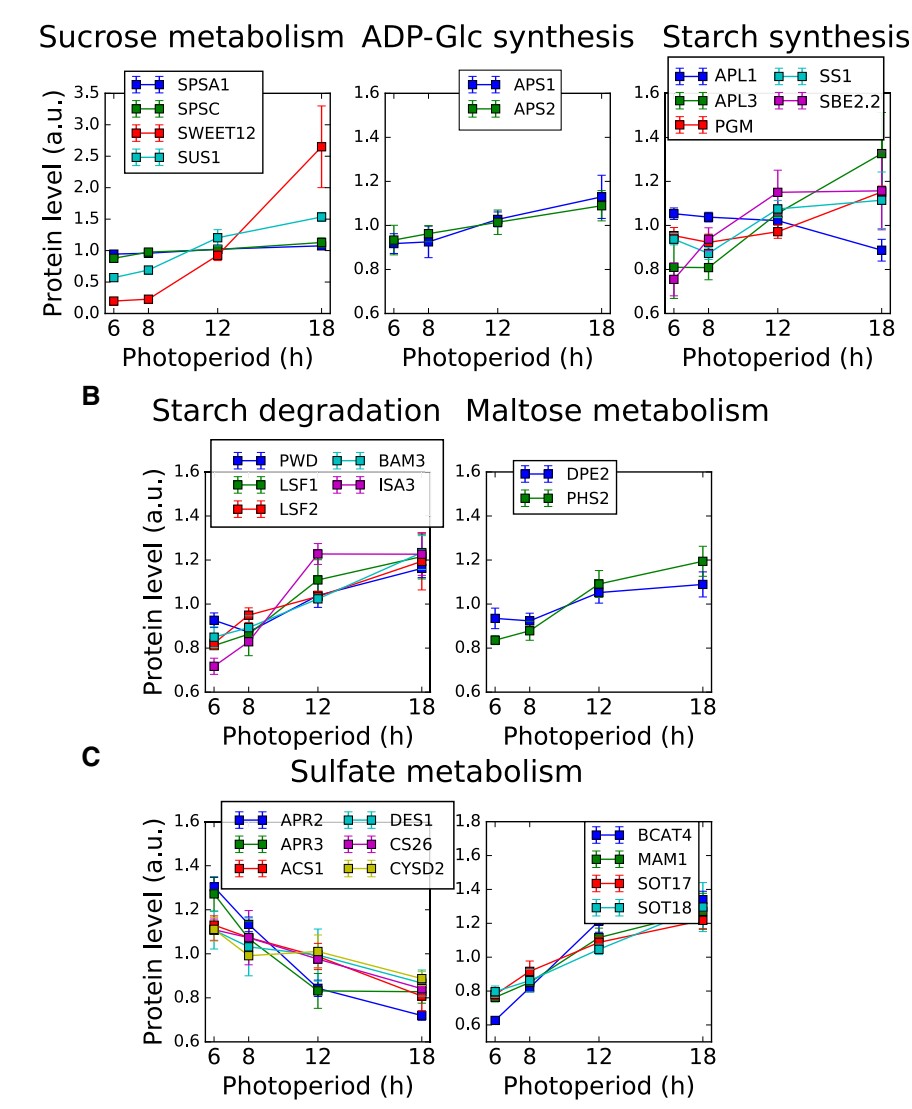

**Figure 4. Photoperiod modulates protein levels of enzymes involved in primary and secondary metabolism.**

A   Significantly changing proteins involved in the partitioning of sugars to sucrose and starch during the day, including sucrose metabolism (sucrose-phosphate synthase, SPSA1, SPSC; bidirectional sugar transporter SWEET12; sucrose synthase, SUS1), ADP-Glc synthesis (glucose-1-phosphate adenylyltransferase small subunits, APS1, APS2) and starch synthesis (glucose-1-phosphate adenylyltransferase large subunits, APL1, APL3; phosphoglucomutase, PGM; starch synthase, SS1; 1,4-alpha-glucan-branching enzyme, SBE2.2).

B   Significantly up-regulated proteins involved in metabolism of starch during the night, including starch degradation (phosphoglucan, water dikinase, PWD; phosphoglucan phosphatase, LSF1, LSF2; beta-amylase, BAM3; iso-amylase, ISA3) and maltose metabolism (4-alpha-glucanotransferase, DPE2; alpha-glucan phosphorylase, PHS2).

C   Significantly down-regulated proteins in sulphate metabolism. Includes 5′-adenylylsulphate reductases (APR2, APR3), cysteine synthases (ACS1, DES1, CS26, CYSD2), methionine aminotransferase (BCAT4), methylthioalkylmalate synthase (MAM1) and cytosolic sulphotransferases (SOT17, SOT18).

Data information: Error bars represent the standard deviation (*n* = 3).

were positively correlated both with transcript changes at ED ($\rho = 0.63$) and EN ($\rho = 0.47$) (Fig EV2A). An overrepresentation analysis of GO terms showed that distinct cellular functions are enriched in transcript-protein pairs that have the same or opposite accumulation pattern, indicating that changes in transcript and protein abundance between photoperiods are highly orchestrated (Table EV8). Next, we identified a subset of transcripts that has no discernible diurnal rhythm in expression (see Materials and Methods for details). We expect the estimates of changes in abundance across photoperiods to be especially accurate for these arrhythmic transcripts because these estimates are not affected by the sparse (two time-points) sampling of expression in each photoperiod. As expected, the correlation of transcript and protein abundance was much stronger for these arrhythmic transcripts both at ED ($\rho = 0.86$) and EN ($\rho = 0.85$; Fig EV2B). Together, these results demonstrate the expected relationship between abundance changes at transcript and protein levels, although this relationship is not strictly followed in all cases, similar to other

species and different experimental conditions (reviewed in Vogel & Marcotte, 2012; Liu *et al*, 2016).

### Light-induced translation provides a mechanism for photoperiodic control of protein expression

Post-transcriptional mechanisms, such as regulation of translation rate, can also play a role in determining protein abundance. In *Arabidopsis*, light induces proteome-wide changes in protein synthesis, as measured by $^{13}CO_2$ labelling (Ishihara *et al*, 2015) or polysome loading (Piques *et al*, 2009; Juntawong & Bailey-Serres, 2012; Liu *et al*, 2012; Pal *et al*, 2013; Missra *et al*, 2015). We considered the effects of this light-dependent translational regulation on the relationship between the transcriptome and proteome in different photoperiods (Fig 5A).

For a gene that is transcriptionally regulated by the circadian clock, the timing of protein synthesis depends on the circadian phase of RNA expression and the light:dark regulation of RNA translation. Coincidence of high RNA transcript levels with a

high rate of translation per transcript (as occurs during the light period) is expected to increase protein synthesis. For a dawn-phased transcript with peak abundance at 2 h after lights-on, for example, high transcript levels coincide with the light interval regardless of photoperiod (Fig 5B). We consider the simple case where the phase of the clock is set by dawn alone, as this is close to the behaviour of the *Arabidopsis* clock (see Discussion, and Edwards *et al*, 2010). An evening-phased transcript, for example peaking 12 h after dawn, has high transcript levels coinciding with the light interval only under long photoperiods (Fig 5B).

This model predicts that differences in the rates of protein synthesis across photoperiods are at least in part due to changes in the coincidence of rhythmic RNA expression with light and the resultant higher rates of translation. We term this mechanism "translational coincidence". Such an interaction between internal (circadian) and external (light:dark) rhythms defines the general "external coincidence" mechanism of photoperiod sensitivity, equivalent to the mechanism proposed to control flowering time (Song *et al*, 2015).

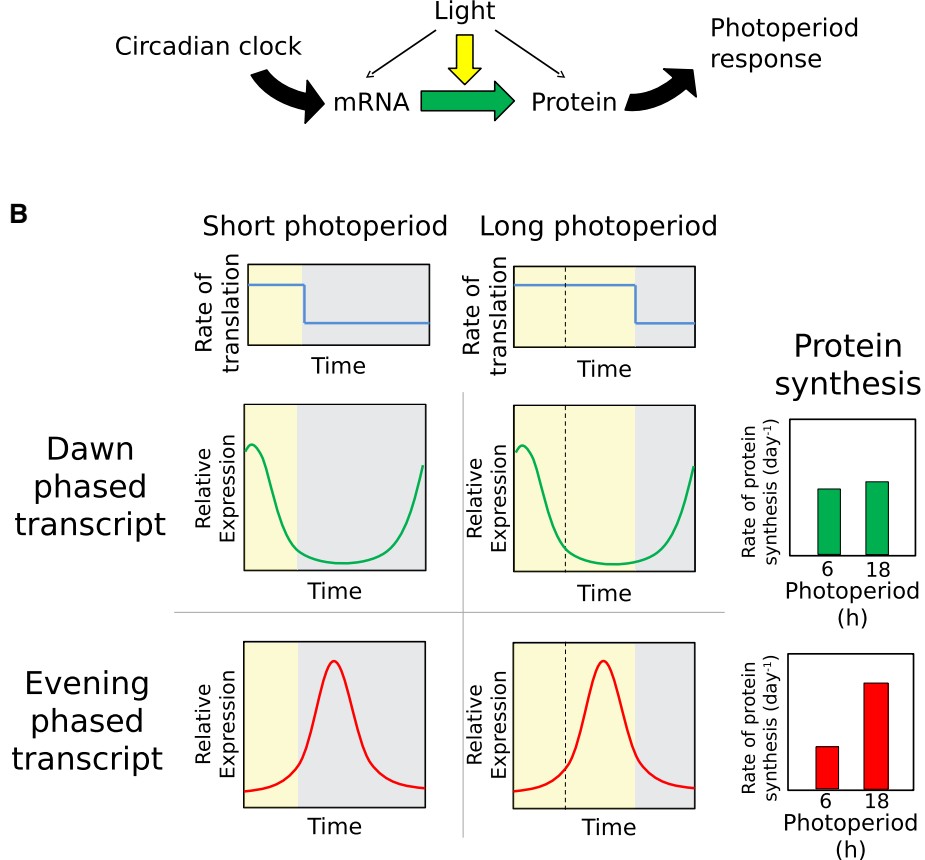

**Figure 5.  Expected effects of the changing coincidence of protein synthesis with transcript.**

A   Schematic of translational coincidence. The combination of circadian regulation of transcript and light stimulation of translation leads to photoperiod responses at the protein level.

B   Idealised representation of translational coincidence for two different transcripts. Light maintains high rates of protein synthesis for longer in longer photoperiods (top panels), which is expected to be without consequence for protein synthesis from dawn-phased transcripts (centre panels), but results in a boost of protein synthesis from transcripts expressed late in the day (bottom panels). The total number of transcript:protein pairs is given in the title of each graph.

## Modulation of photoperiod-dependent protein expression is explained by translational coincidence

If circadian-controlled gene expression contributes to changes in protein levels across photoperiods, we expect an overrepresentation of circadian-controlled genes in the set of differentially regulated proteins. Comparing the consensus set of circadian-controlled transcripts reported in Covington *et al* (2008) to the photoperiod-regulated proteins in our present data set, this is indeed the case ($P < 0.001$; hypergeometric test). We therefore tested the more specific predictions of the translational coincidence hypothesis, relating late-peaking transcripts to protein accumulation in long photoperiods, before including the predicted effects of translational regulation.

Starting first from the transcript regulation, we examined the timing of transcript expression for proteins identified as up-regulated and down-regulated in long photoperiods. Full diel time series data were previously acquired in a 12-h photoperiod (Bläsing *et al*, 2005). This dataset is particularly suitable in this context because it reveals transcript dynamics in soil-grown adult rosettes of a similar age (35 days) to those used in our experiments (30 days). We augmented the dataset with the peak-phase annotation calculated in the DIURNAL database by waveform interpolation (Mockler *et al*, 2007; Michael *et al*, 2008a), binned into 2 h windows. Among the sets of evening-phased transcripts, proteins that accumulated to higher levels in long photoperiods were overrepresented. Such proteins were under-represented among sets of dawn-phased transcripts (Fig 6A). The converse was true for proteins that had lower abundances in long photoperiods (Fig 6A). These observations are consistent with the translational coincidence hypothesis. In particular, we note that the lower abundance of proteins with dawn-phased transcripts in long photoperiods follows from the increased rate of dilution of all proteins by increased growth rate in long photoperiods (see below and Models and methods: Mathematical model of translational coincidence for details).

Starting next from the protein changes, we calculated changes in protein level between short (6 h) and long (18 h) photoperiods for all proteins (i.e. not only those identified as changing significantly in protein level). These were binned by the phase of transcript expression, for the subset of 547 proteins with transcripts displaying rhythms with peak/mean amplitude > 1.5 (Fig 6B). This showed a clear pattern of responses across the diurnal cycle, with dawn-phased transcripts tending to have proteins with lower abundance and evening-phased transcripts tending to have higher abundance proteins, with a progressive response across all four photoperiods (Fig 6C and D). This association was also observed in the dataset of Baerenfaller *et al* (2015) of *Arabidopsis* leaf 6 protein levels in 8-h and 16-h photoperiods (Fig EV3). Importantly, similar patterns of photoperiod sensitivity were observed in the protein abundances measured at either EN or ED time point, and across different leaf developmental stages (Baerenfaller *et al*, 2015). This consistency between datasets confirms that the observed phase relationship does not result from the sampling times in our dataset, either during development or at the time of day.

We next simulated translational coincidence in a quantitative model, using both transcript dynamics (Bläsing *et al*, 2005) and changes in bulk protein synthesis rates measured by $^{13}CO_2$ labelling (Pal *et al*, 2013) to predict changes in protein synthesis, and consequently abundance, across photoperiods (Fig 6E). This model accounts for changes in protein synthesis depending on light and mRNA abundance. Changes in protein synthesis then result in changes in abundance after accounting for changes in the rate of bulk protein synthesis with photoperiod (i.e. accounting for changes in growth with photoperiod) (see Models and methods: Mathematical model of translational coincidence for details). Similar approaches have previously been applied to model isotope labelling of protein in growing tissues (Ishihara *et al*, 2015). Protein abundance in a specified photoperiod is then given by:

$$P_{normalised} = \frac{R \int_{t=0}^{t_{dusk}} m(t) + \int_{t=t_{dusk}}^{24} m(t)}{(R-1)t_{dusk} + 24}$$

Where $P_{normalised}$ is the predicted, normalised protein abundance, $R$ is the ratio of translation in the light compared to the dark, $m(t)$ is the mRNA abundance as a function of time, and $t_{dusk}$ is the time of dusk (see Materials and Methods: Translational coincidence for detailed model description). We predicted changes in protein levels between 6-h and 18-h photoperiods for 251 proteins with a high-amplitude rhythm in their transcript (> 1.7-fold difference between peak level and mean). Our measured protein levels for these photoperiods, which were not used to build the model, can now test its predictions. There was a highly significant agreement (Pearson's $R = 0.41, P < 10^{-10}$) between the model prediction and the measured changes in protein levels, with the model also quantitatively matching the proportional relationship (gradient of slope = 0.75) (Fig 6F). This result demonstrates that the effect of photoperiod on protein accumulation quantitatively matched what we expect from the translational coincidence mechanism, which follows from the rhythmic transcript dynamics and protein synthesis rates.

While the translational coincidence model captured this important trend in the whole dataset, individual proteins varied widely, as quantified by the correlation between model predictions and measurements (Pearson's $R = 0.41$). Several factors are likely to contribute to this variation, including transcript-specific differences in the sensitivity of translation to light, protein-specific changes in turnover with photoperiod, photoperiod-specific transcriptional regulation in response to changes in sugar- or light-signalling (Flis *et al*, 2016), and experimental error in measurements of transcript and protein abundances.

In order to adjust for potentially confounding effects of transcriptional regulation, we removed proteins from consideration according to three complementary criteria aimed at identifying transcripts under consistent regulation by the circadian clock. First, we compared the transcriptome time series dataset from Bläsing *et al* (2005) to a dataset based on EN and ED samples in 4-, 6-, 8-, 12- and 18-h photoperiods from Flis *et al* (2016). This dataset was processed by averaging the EN samples to represent ZT0, and taking successive ED time-points to represent ZT4, 6, 8, 12 and 18. A high degree of correlation between these datasets then indicates an underlying (putatively circadian) rhythm that is robust to changes in photoperiod. As confirmation that this criterion identifies transcripts under strong circadian regulation, we note that as expected, the core circadian clock genes *CCA1*, *PRR7*, *ELF3* and *TOC1* all pass this test (Fig EV4). Taking a threshold of Pearson's correlation of 0.75 reduced the set of proteins considered from 547 to 341. The pattern of photoperiod response remained the same after this filtering (Fig EV5A).

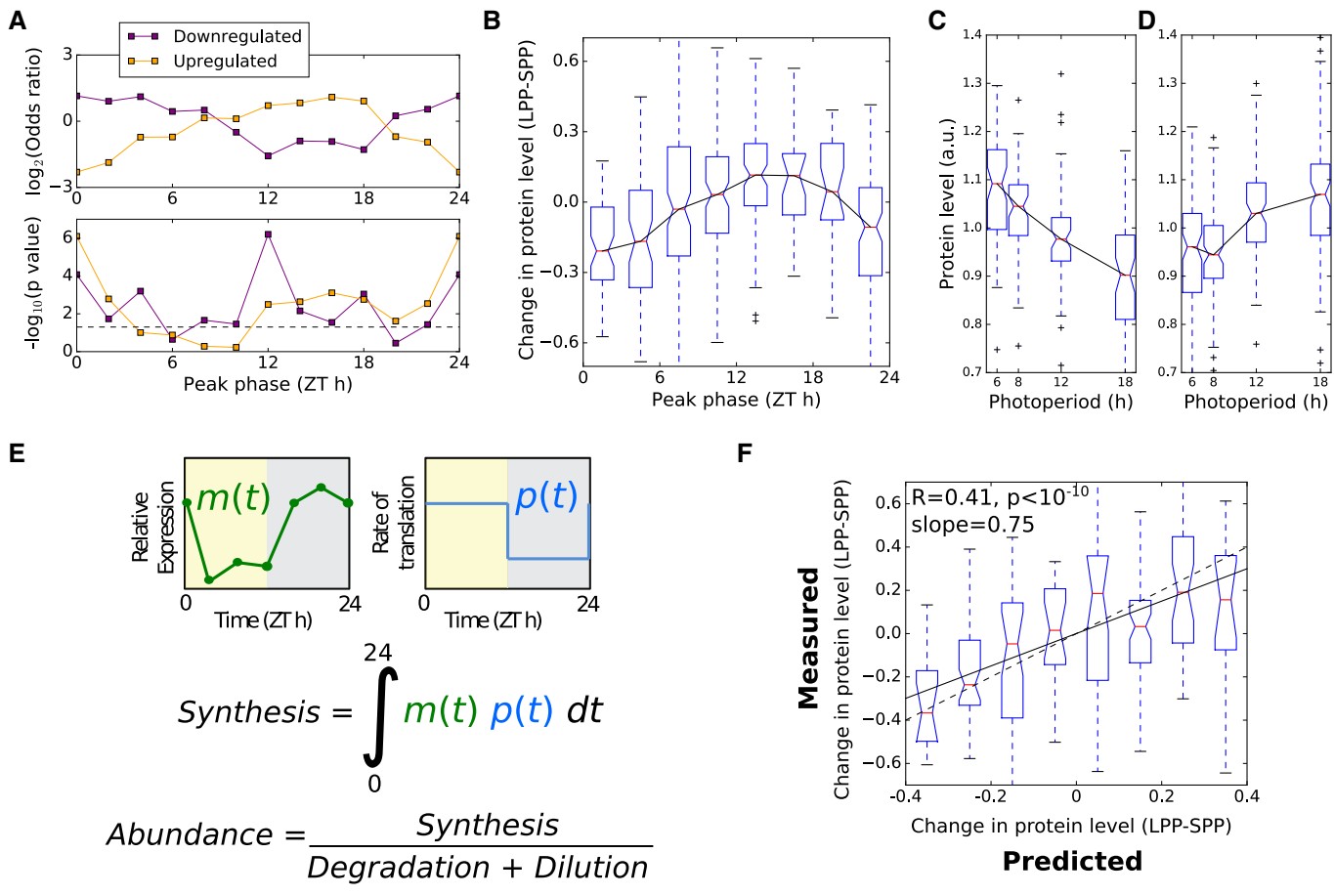

**Figure 6.  Evaluating circadian control of protein changes with photoperiod.**

A   Phase enrichment of proteins identified as significantly up- and down-regulated in long photoperiods, evaluated by Fisher's exact test, with transcripts grouped by phase in two-hour intervals according to phase of expression in the microarray timecourse dataset of Bläsing *et al* (2005).

B   Change in protein level between short (6 h) and long (18 h) photoperiods (LPP-SPP), grouped according to the peak phase of transcript expression.

C   Progressive changes in protein abundance across photoperiods for proteins whose transcripts peak in expression between ZT0 and ZT2 in the dataset of Bläsing *et al* (2005).

D   As in (C), for proteins whose transcripts peak between ZT12 and ZT14.

E   Schematic of a simple model of protein synthesis, using measured mRNA (*m*) and translation (*p*) input data.

F   Comparison of model to data, for changes between 6-h and 18-h photoperiods (LPP-SPP) for the 251 proteins with rhythms in RNA abundance with amplitude > 1.7. Changes are plotted as differences between photoperiods, normalised to the mean. The dashed line indicates the case where model predictions match measured values. The solid line indicates the linear fit to the plotted data.

Data information: Boxes span the interquartile range. Whiskers span 1.5 times the interquartile range.

Second, we compared expression dynamics in continuous light to dynamics in light:dark cycles. Genes that are predominantly regulated by the circadian clock are expected to have similar rhythms of transcript accumulation in both conditions. We therefore identified transcripts with circadian-dominant accumulation dynamics by calculating the Pearson's correlation coefficient between the (Bläsing *et al*, 2005) diel and a circadian transcriptome time series (Covington & Harmer, 2007), again filtering out transcripts with correlation coefficients < 0.75. As above, we note that core clock genes pass this test (Fig EV4). This filter reduced the number of proteins considered from 547 to 142. Figure EV5B shows that the qualitative distribution of protein level changes remains similar after this filtering.

Third, we compared mean expression levels in short (8 h) and long (16 h) photoperiods, using the transcriptome time series dataset of (Michael *et al*, 2008b) (data from 2-week-old seedlings,

Landsberg (Ler) accession). We identified transcripts with photoperiod-dependent mean expression levels in this dataset by calculating the change in expression, and filtering out transcripts with a > 0.3-fold change in expression. This filter reduced the number of proteins considered from 547 to 378. The pattern of photoperiod response remained the same after this filtering (Fig EV5C). Furthermore, this pattern remained after combining all three filters, reducing the total number of proteins considered further to 104 (Fig EV5D). The transcripts in this set displayed no discernible relationship between phase of expression and change in abundance in the dataset of (Michael *et al*, 2008b; Fig EV5E). The translational coincidence model maintained a good fit to the data for this subset (Pearson's *R* = 0.5; Fig EV5F).

The reduced set of transcripts remaining after filtering is too small to draw conclusions using enrichment analyses. However, specific examples illustrate the potential effects of a translational

coincidence mechanism on plant physiology in changing photoperiods. Two examples of dawn-phased transcripts with decreases in protein levels in longer photoperiods are GENOMES UNCOUPLED 4 (GUN4) and GUN5 (Fig EV6A). These proteins are involved in chlorophyll biosynthesis, and their transcripts are robustly phased to dawn by the circadian clock. Two examples of evening-phased transcripts that increase in protein level with photoperiod are ALPHA-GLUCAN PHOSPHORYLASE 2 (PHS2) and ISO-AMYLASE 3 (ISA3), which are involved in starch turnover (Fig EV6B).

In summary, our results are consistent with the translational coincidence hypothesis, whereby protein levels are influenced by the coordinated timing of transcript expression and light-regulated protein synthesis. Translational coincidence may be an important regulatory mechanism for slowly turning-over proteins with transcripts that are regulated by the circadian clock. The mechanism changes protein abundance in response to photoperiod, without photoperiodic regulation of transcript abundance. In coupling *daily* RNA rhythms to *seasonal* physiology, it supports broadly the same operating principle that has been highly adapted in the specialised, photoperiodic flowering mechanism (see Discussion).

### Translational coincidence as a general mechanism of photoperiod sensitivity in phototrophs

Translational coincidence depends on only two key parameters, faster protein synthesis in the light and circadian control of gene expression, which might operate in many phototrophic organisms. We therefore examined existing proteome and transcriptome datasets for the green alga *Ostreococcus tauri* and the cyanobacteria *Cyanothece* ATCC51142 (proteome) and *Synechococcus elongatus* PCC7942 (transcriptome). These datasets do not characterise the photoperiodic responses of the proteomes and transcriptomes of these species. Nevertheless, they allow assessment of whether light-induced protein synthesis and circadian regulation of gene expression are observed in these cases.

Quantitative proteome time courses across light:dark cycles using stable isotope labelling in *O. tauri* (Martin *et al*, 2012) and *Cyanothece* (Aryal *et al*, 2011) allow inference of relative rates of protein synthesis in the light and dark on a protein-by-protein basis, analogous to calculations performed for *Arabidopsis* (Pal *et al*, 2013; Ishihara *et al*, 2015). The median relative rates of isotope incorporation in the light compared to the dark were 4.7 for *O. tauri* and 3.2 for *Cyanothece* (Fig 7A). Protein synthesis and biomass accumulation measurements show similar patterns in *Synechococcus* spp. (Glover & Smith, 1988), the unicellular red alga *Cyanidioschyzon merolae* (Miyagishima *et al*, 2014), the cyanobacterium *Arthrospira plantensis* (Matallana-Surget *et al*, 2014) and the marine diatom *Thalassiosira pseudonana* (Ashworth *et al*, 2013). Diel and circadian regulation of the transcriptome has also been tested. In *O. tauri*, about 80% of transcripts change across a diel cycle (Monnier *et al*, 2010), while in *Synechococcus* about 30% of transcripts cycle with circadian rhythms (Ito *et al*, 2009; Fig 7C). Thus, both light-induced protein synthesis and diel regulation of gene expression are observed in diverse species.

In *Arabidopsis*, many proteins have a half-life of several days (Li *et al*, 2017) and the half-life of total protein was estimated as 3–4 days (Ishihara *et al*, 2015). Because of these low relative protein turnover rates, diel transcript cycling does not generally translate to diel dynamics at the protein level (Piques *et al*, 2009; Baerenfaller *et al*, 2012; Stitt and Gibon, 2014). We examined estimated rates of protein turnover in *Arabidopsis*, *O. tauri* and *Cyanothece* based on isotope labelling and quantitative proteome data (Aryal *et al*, 2011; Martin *et al*, 2012; Li *et al*, 2017; see Materials and Methods: Inference of protein synthesis and degradation rates in Ostreococcus and Cyanothece for details). The distributions of calculated rates of protein turnover in Fig 7B show that low rates of degradation are found in all three organisms. Furthermore, only about 5% of measured proteins in *Synechococcus* have diel dynamics, which is also consistent with a slow turnover of most measured proteins (Guerreiro *et al*, 2014). Translational coincidence would therefore cause a slow response of protein levels to changes in photoperiod over several days, potentially matching the gradual change of photoperiod in natural environments.

The change in the phase of the circadian clock in response to changing photoperiods is a dynamic property termed "dusk sensitivity" (Edwards *et al*, 2010). The circadian clock in *Arabidopsis* primarily tracks dawn across photoperiods (it has low dusk sensitivity, as in Fig 4, see also Flis *et al*, 2016). Thus *Arabidopsis* rhythms have a consistent phase of entrainment, relative to dawn, across a wide range of photoperiods (Edwards *et al*, 2010; Flis *et al*, 2016). However, circadian clocks in other species track dusk (e.g. *Ipomoea nil*; Heide *et al*, 1988) or show an intermediate behaviour (e.g. "noon-tracking" clocks, as in *Neurospora crassa*; Tan *et al*, 2004). These distinct circadian behaviours are illustrated in Fig EV7 for a transcript that peaks at dusk in 12/12 light/dark conditions. Clocks with these properties are predicted to alter the protein response to photoperiod. A dawn-tracking clock allows up-regulation of a protein with a dusk-peaking transcript under long photoperiods, as in *Arabidopsis*, whereas a noon-tracking clock yields photoperiod-insensitivity, and a dusk-tracking clock yields down-regulation of protein levels with increasing photoperiod.

The pre-conditions for translational coincidence are present in a wide variety of phototrophic organisms, suggesting that this mechanism might affect protein levels very broadly. However, the translational coincidence mechanism is flexible. The details of the photoperiod response can be tuned by the rhythmic expression profile of individual RNAs, by the light-sensitivity of the translation rate and globally by the dusk sensitivity of the circadian clock.

## Discussion

Many aspects of plant development, physiology and metabolism have been demonstrated to respond to changes in photoperiod. Here, we quantified the response of the *Arabidopsis* proteome to changing photoperiods, providing a broad view of photoperiodic responses across many pathways and processes. This revealed several processes that are regulated by photoperiod, ranging from photosynthesis and primary metabolism to secondary metabolism and growth. Furthermore, we made a new, mechanistic link from the light dependence of protein synthesis and rhythmic transcript regulation to the observed responses of protein abundance to photoperiod. This has implications for our understanding of photoperiod responses in plants and other phototrophic species. Translational coincidence can explain how plants adjust their

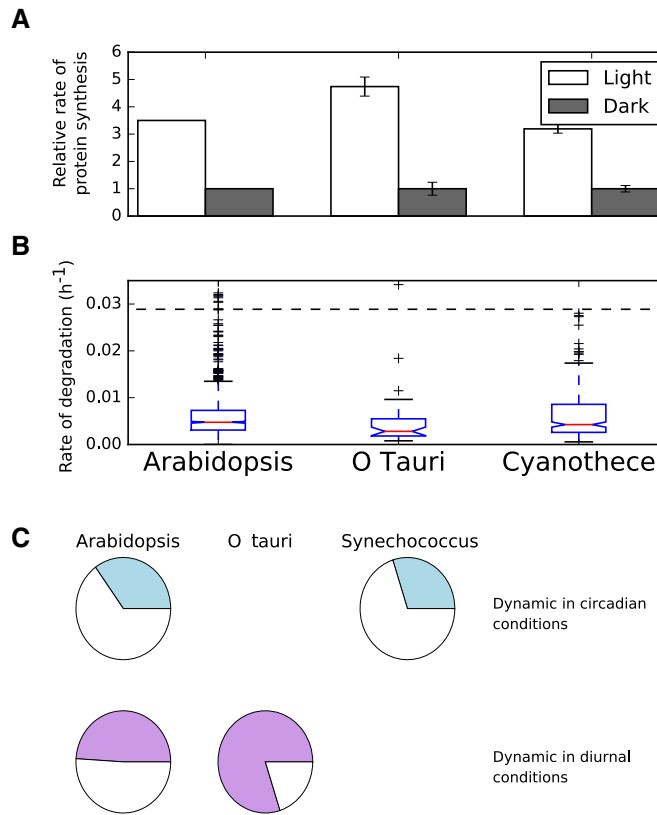

**Figure 7.  Ingredients of translational coincidence in diverse photoautotrophic organisms.**

A   Light-stimulated protein synthesis. Relative rates of protein synthesis in the light compared to the dark have been reported in *Arabidopsis* (Pal *et al*, 2013) and were inferred from quantitative proteomics stable isotope labelling datasets for Ostreococcus (Martin *et al*, 2012) and Cyanothece (Aryal *et al*, 2011) (see Materials and Methods for details). Error bars represent the standard error ($n = 40$ for Ostreococcus data, $n = 750$ for Cyanothece data).

B   Slow rates of protein turnover. The dashed line represents a half-life of 1 day. Protein degradation rates have been reported for *Arabidopsis* (Li *et al*, 2017) and were inferred from quantitative proteomics data for Ostreococcus and Cyanothece, as in (A) (see Materials and Methods for details). Boxes span the interquartile range. Whiskers span 1.5 times the interquartile range.

C   Diurnal and circadian dynamics in gene expression. Shaded areas represent the fraction of the transcriptome estimated to be dynamic in circadian (top row) and diurnal (bottom row) conditions.

proteome to prevailing photoperiods, optimising their metabolism and growth. Widespread circadian regulation of RNAs might provide a selective advantage by this mechanism, even if their cognate proteins are too stable to show daily rhythms.

### Coordinated changes in response to photoperiod

Our quantitative analysis of 4,344 *Arabidopsis* proteins in different photoperiods revealed highly coordinated changes in the abundance of proteins across a wide range of metabolic pathways. Proteins with related functions not only tend to change in abundance in the same direction but also within very narrow FC windows. A previous study using the same plant material as in our work reported here

showed that plants in the 18-h photoperiod differ strongly in their phenotype and metabolic state compared to shorter photoperiods, including changes in leaf morphology, pattern of starch accumulation and degradation, and carbon-conversion efficiency (Sulpice *et al*, 2014). In general, plant growth in long photoperiods is no longer carbon-limited (Baerenfaller *et al*, 2015). Changes in protein abundance are mostly gradual between photoperiods, although some proteins have abrupt increases or decreases in abundance between neighbouring photoperiods. Together, the proteome changes not only across the photoperiod range where growth is increasing in response to an increasing fixed carbon supply but also in the range where the fixed carbon supply exceeds the requirement for growth. Similarly, the end-of-night and end-of-day transcriptomes show progressive changes across the entire range from a 4-h to an 18-h photoperiod (Flis *et al*, 2016).

Several metabolic pathways in plants are preferentially active during the light or the dark period. The proteome in different photoperiods reflects the adjustment to the increasing ratio between the light and dark phase of the diurnal cycle in longer photoperiods. Longer photoperiods show a concerted down-regulation of metabolic pathways that are predominantly active in the light including fatty acid biosynthesis, the MEP pathway and chlorophyll biosynthesis (Bao *et al*, 2000; Eckhardt *et al*, 2004; MongéLard *et al*, 2011). In contrast, enzymes involved in fatty acid degradation were more abundant in longer photoperiods. Similar to other oxidative processes, the degradation of fatty acids requires $NAD^+$. Considering the rapid formation of NADPH in photosynthesis and rapid conversion of $NAD^+$ to NADH during photorespiration, it is plausible that fatty acid oxidation occurs preferentially in the dark. Plants might therefore up-regulate fatty acid degrading enzymes in longer photoperiods to compensate for the shorter dark period with an increased flux through this pathway.

Ribosomes are among the most abundant protein complexes of a plant cell. In *Arabidopsis*, higher protein synthesis rates were observed in the light compared to the dark period (Piques *et al*, 2009; Juntawong & Bailey-Serres, 2012; Liu *et al*, 2012; Pal *et al*, 2013; Ishihara *et al*, 2015; Missra *et al*, 2015). This might reflect a strategy for optimal use of fixed carbon, since protein synthesis during the night requires sequestration of fixed carbon during the day, which entails additional energetic costs (Pal *et al*, 2013). More fixed carbon is available for metabolism in long photoperiods than short photoperiods. In agreement, polysome loading was decreased in the dark compared to the light period in short photoperiods. This difference became progressively smaller as the photoperiod was lengthened and polysome loading was similar in the day and the night in long photoperiods (Sulpice *et al*, 2014). This reflects the higher rates of starch degradation and higher levels of sugars during the night in long compared to short photoperiods. Hence, in longer photoperiods, plants can use their translational machinery over a longer period of time per diurnal cycle and a lower translation capacity might be sufficient to establish and maintain the proteome of each cell. A decrease in ribosomal protein abundance was also observed across leaf development (Baerenfaller *et al*, 2012). Mature leaves require their translational machinery mainly for maintenance while young leaves have to fully establish their proteome and might therefore need a higher capacity for protein synthesis. In both scenarios, the change in photoperiod length and leaf development, the

down-regulation of ribosome abundance may reflect an optimised use of energy, nitrogen and carbon resources required to establish and maintain a set of highly abundant proteins.

Interestingly, both RPS6 isoforms that are essential for the 40S ribosomal subunit (Creff *et al*, 2010) were up-regulated in longer photoperiods. Moreover, RPS6 is a highly conserved target of TOR kinase and thus may integrate signals of the nutritional and light energy status with the regulation of growth and life span in *Arabidopsis* plants (Ren *et al*, 2012). The photoperiod-dependent increase in RPS6 abundance in the context of general down-regulation of the translational machinery indicates that bulk translation capacity is complemented by yet unknown processes to modulate protein synthesis in different photoperiods.

Other changes in protein abundance between photoperiods cannot be attributed to a longer or shorter window of activity of specific pathways, but reflect a re-programming of plant metabolism to optimise the efficiency of carbon use. For example, orchestrated up-regulation of sugar- and starch-related enzymatic pathways indicates that plants in longer photoperiods have a highly active primary carbon metabolism, including a higher capacity for starch degradation. There is an increase in abundance for many proteins in electron transport, the Calvin–Benson cycle, sucrose and starch synthesis, and the TCA cycle, indicating a higher capacity for carbon assimilation and use. The increase in starch degradation and TCA cycle enzymes could support increased fluxes in respiration metabolism to provide energy and reducing equivalents for biosynthetic reactions and sugar intermediates for rapid growth. Moreover, the strong up-regulation of sucrose export proteins in longer photoperiods, including the SWEET12 protein, indicates that source leaves might have an increased capacity for sucrose export to support growth in sink organs. Compared to short photoperiods, in long photoperiods plants synthesise less starch and therefore export sucrose more rapidly in the light period, while they degrade starch and export sucrose more rapidly during the night (Sulpice *et al*, 2014; Mengin *et al*, 2017). This is consistent with the increased abundance of many proteins in the starch degradation pathway in long photoperiods. The low abundance of these proteins in short photoperiods would be expected to restrict the rate of starch degradation when starch must be conserved until dawn (Sulpice *et al*, 2014; Baerenfaller *et al*, 2015).

Consistent with the general up-regulation of enzymes in primary carbon metabolism, sucrose, sucrose-6-phosphate and glucose-6-phosphate levels increase with photoperiod length (Sulpice *et al*, 2014). This links primary carbon metabolism to the synthesis of sulphur-containing and defence-related glucosinolates, which is positively regulated by sugars on the transcript level (Gigolashvili *et al*, 2007; Guo *et al*, 2013; Miao *et al*, 2013; Flis *et al*, 2016). We found that the enzymes in the glucosinolate pathway accumulated to higher levels in longer photoperiods. The levels of these enzymes also increase during leaf development (Baerenfaller *et al*, 2012, 2015). In both scenarios—increased photoperiod length and later stages of leaf development—plants invest more resources into the synthesis of defence-related compounds when available energy and assimilated carbon are less restricted (del Carmen Martínez-Ballesta *et al*, 2013). Similarly, the increased levels of isoprenoid biosynthesis enzymes in the MVA pathway similarly supports increased synthesis of defence-related terpenoids in longer photoperiods (Vranová *et al*, 2013).

Several photoperiod-dependent changes in protein abundance could alter protein complex composition rather than affecting the regulation of entire enzymatic pathways. AGPase, which catalyses the first committed and rate-limiting step of starch synthesis, represents such an example. The AGPase complex integrates signals of cellular carbon metabolism, thereby regulating the partitioning between carbon storage, export and utilisation (Orzechowski, 2008). The heterotetrameric complex of two large APL1 and two small APS1 subunits is responsible for 95% of the AGPase activity in the *Arabidopsis* rosette (Wang *et al*, 1997). APL1 abundance is decreased and APL3 increased in longer photoperiods while APS1 subunits were unchanged, indicating that APL3 might at least partially substitute APL1 in the AGPase complex in longer photoperiods. APL3 can be induced by exogenous provision of sugars and functionally complements APL1-deficient mutants, suggesting that APL1 and APL3 confer different regulatory properties to the AGPase complex (Wingler *et al*, 2000; Fritzius, 2001).

The PSII supercomplex in the photosynthesis electron transport chain is another example of photoperiod-dependent changes in complex composition. We observed a general increase in abundance for components of all electron transport chain complexes, including PSII, in longer photoperiods. In line with reports showing that the minor light harvesting antenna CP29 is present in a 1:1 ration to the PSII core complex independent of the light conditions (Ballottari *et al*, 2007; Bielczynski *et al*, 2016), we found that two CP29 isoforms (LHCB4.1 and LHCB4.3) were also up-regulated. However, several isoforms of the major PSII LHCII antenna complex decreased in abundance. This could indicate a shift in stoichiometry between core proteins and the LHCII antenna in the PSII supercomplex. Such shifts in stoichiometry were observed in *Arabidopsis* during acclimation to different light intensities (Bielczynski *et al*, 2016). Under natural light regimes, a longer photoperiod at a given geographical location is likely to be associated with higher peak light intensities. This might provide an explanation for the trend to increased electron transport capacity but decreased light antennae in long photoperiod-grown plants, as well as the trend to increased abundance of proteins in the downstream reactions in photosynthetic carbon metabolism. However, LHCII proteins can also be present as monomers in the thylakoid membrane and this PSII-independent fraction was shown to change during light acclimation (Wientjes *et al*, 2013; Bielczynski *et al*, 2016). Thus, a decrease in the abundance of the monomeric LHCII fraction could also explain the diametrical change in abundance we observed between LCHII and PSII core proteins in longer photoperiods.

## Translational regulation contributes to protein responses to photoperiod

Rates of protein synthesis are typically higher in the light, when they are driven by the energy and fixed carbon generated by photosynthesis, than in the dark when they rely on the mobilisation of carbon reserves. Transcripts that peak early during the clock cycle will be efficiently translated in both short and long photoperiods. In contrast, transcripts that peak in the middle of the clock cycle will be efficiently translated in long photoperiod but not in short photoperiods. We term this mechanism of photoperiod response "translational coincidence", and analysis of our proteomics dataset demonstrates its role in mediating photoperiod responses in *Arabidopsis*.

The action of light on translation is probably indirect. In the plastid, where up to half of the protein synthesis occurs in a leaf, the provision of energy and especially ATP by photosynthesis may underlie the strong light dependence of protein synthesis (Marín-Navarro *et al*, 2007; Pal *et al*, 2013). In the cytosol, it is less clear that light impacts directly on the energy status, which in the dark is also maintained at a high level by oxidative phosphorylation (Stitt *et al*, 1982; Gardeström & Wigge, 1988; see Stitt *et al*, 2010 for a review). Stimulation of cytosolic protein synthesis in the light may be due to the rise in sugar levels (Pal *et al*, 2013). Therefore, it can be questioned if translational coincidence will be robust against fluctuations in light intensity that affect the rate of photosynthesis and the supply of energy and carbon in the light. While more analyses is required to establish this, it is likely to be robust because a decreased rate of photosynthesis in low light affects not only sucrose synthesis but also starch accumulation and the carbon status in the following night, during which remobilisation of starch is required for maintenance and repair (Pilkington *et al*, 2015; Mengin *et al*, 2017). Thus, while low light will decrease protein synthesis in the light, it is likely to result in an even greater decrease of the low rate in the following night. Further, as the photoperiod lengthens, the increasing rate of starch degradation allows higher levels of sugars to be maintained at night, which could support increased polysome loading during the night (Sulpice *et al*, 2014). This trend will reinforce the proposed translational coincidence mechanism; translation of transcripts that peak in the middle of the circadian cycle will be strongly restricted in short photoperiods, whereas they may still be translated at relatively high rates even after dusk in long photoperiods.

### The circadian clock tunes protein responses to photoperiod

The role of the circadian clock in mediating photoperiodic changes in protein expression suggests an explanation for a longstanding paradox. Diel rhythms in transcript level often do not lead to diel rhythms in protein level (Gibon *et al*, 2004; Lu *et al*, 2005; Baerenfaller *et al*, 2012). Therefore, what is the physiological significance of the pervasive, rhythmic transcript regulation, if any? It may be that the functional properties of stable proteins that are newly synthesised (when RNA levels are high) differ from the existing bulk pool (Busheva *et al*, 1991), though this seems unlikely to be a general case. The translational coincidence mechanism suggests that diel RNA rhythms might tune the levels of many proteins on a seasonal timescale, rather than within a single day.

It is an open question whether the photoperiod responses we observe are adaptive for these different conditions, or are merely a tolerated consequence of growth in different photoperiods. However, as indicated above, the changes in abundance of proteins involved in carbon metabolism, secondary metabolism and the translational machinery certainly have the potential to contribute to the change in metabolite levels and fluxes in different photoperiods. A similar interplay of clock and translation may also be relevant in other systems where protein levels do not change significantly over the course of a day. For example, macromolecule biosynthesis exhibits a strong diel rhythm in the mouse liver (Atger *et al*, 2015), though the proteome shows only weak diel rhythms (Mauvoisin *et al*, 2014). Similar questions have arisen in the study of microbial organisms, in which changes in protein synthesis rates also have

widespread effects. For example, changes in ribosome loading at higher growth rates are known to affect the proteome composition in *Bacillus subtilis* (Borkowski *et al*, 2016).

### Translational coincidence as a general mechanism of photoperiodic regulation

Changes in photoperiod place significant demands on plant physiology and growth. Given that the demands of growth in varying photoperiods are likely to be similar for plant species, we might expect similar changes in proteome expression in these species, especially in core processes such as primary metabolism. However, *Arabidopsis* in the laboratory can grow in photoperiods as short as 3 h (Piques *et al*, 2009) and in the field different accessions are found across an especially large range of latitudes, ranging from the north of Scandinavia to the Cape Verde Islands (Koornneef *et al*, 2004), from a possible origin in Africa (Durvasula *et al*, 2017). Thus, *Arabidopsis* may be especially suited to respond to the prevailing photoperiod.

Besides identifying photoperiod-sensitive processes that may be general to plant life, we have also identified a general mechanism of response to photoperiod, termed translational coincidence. The requirements for translational coincidence are simple—light-stimulated translation, and circadian regulation of transcription. As illustrated by several examples, these are general properties of phototrophic life. If other temporally restricted factors regulate translation, rather than light, then translational coincidence might occur in further taxa.

## Materials and Methods

### Plant material and growth conditions

The same plant material used for transcriptome analysis in (Flis *et al*, 2016) was the basis of our proteome study. Briefly, *Arabidopsis thaliana* Col-0 plants were grown on GS 90 soil mixed in a ratio 2:1 (v/v) with vermiculite. Plants were grown for 1 week in a 16 h light (250 µmol m$^{-2}$ s$^{-1}$, 20°C)/8 h dark (6°C) regime followed by an 8 h light (160 µmol m$^{-2}$ s$^{-1}$, 20°C)/16 h dark (16°C) regime for 1 week. Plants were then replanted with five seedlings per pot, transferred for 1 week to growth cabinets with an 8-h photoperiod (160 µmol m$^{-2}$ s$^{-1}$, 20°C throughout the day/night cycle) and then distributed into small growth cabinets with an 18-, 12-, 8- or 6-h photoperiod (160 µmol m$^{-2}$ s$^{-1}$ and 20/18°C in the day/night). This growth protocol was used to decrease differences in size between plants at the time of harvest and to prevent an early transition to flowering that would otherwise occur if plants were grown from germination in long photoperiods. Plant material was harvested 9 days after transfer, at the end of the day (end-of-day samples were taken prior to lights switching off). Plant material was homogenised using a Ball-Mill (Retch, Germany). Approximately 50 mg of material per sample was aliquoted into 2-ml Eppendorf tubes while frozen and distributed for analysis to consortium partners in three biological and two technical replicates. The use of three biological replicates has previously been found suitable for quantifying proteome changes in *Arabidopsis* (Baerenfaller *et al*, 2012).

## Protein extraction and digestion

Frozen plant material was suspended in 100 μl SDS extraction medium [4% w/v SDS, 40 mM Tris, 60 μl ml$^{-1}$ protease inhibitor cocktail (Roche)] and mixed vigorously. The extract was cleared by centrifuged for 10 min at 16,000 *g* followed by ultracentrifugation at 100,000 *g* for 45 min. The resulting supernatant was diluted 4:1 (v/v) in Laemmli sample buffer and incubated at 65°C for 5 min. For each sample, 400 μg protein was subjected to electrophoresis overnight on a 10% SDS–polyacrylamide gel at 60 V. Samples were loaded randomised on the gels to minimise positional effects. Gels were stained in Coomassie Blue solution (20% v/v methanol, 10% v/v acetic acid, 0.1% m/v Coomassie Brilliant Blue R) for 45 min then de-stained twice in 10% v/v methanol, 5% v/v acetic acid for 1 h at room temperature. Each lane of the gel was cut into 7 fractions and transferred to a 96-deep well plate. Gel pieces were fully de-stained by three rounds of 50% v/v methanol, 100 mM ammonium bicarbonate, incubating each time for 1 h at 37°C. In-gel digestion of proteins using trypsin was performed as previously reported (Shevchenko *et al*, 1996). Volumes of solutions were adjusted to ensure that the gel pieces were fully covered during the reduction, alkylation and washing steps. Following in-gel tryptic digestion peptides were purified by reversed-phase chromatography on Finisterre C18 SPE columns (Teknokroma, Barcelona, Spain) and dried in a vacuum centrifuge at 45°C.

## Mass spectrometry analysis

Peptides were re-suspended in 40 μl 3% v/v acetonitrile, 0.1% v/v formic acid. Measurements were performed on a LTQ-Orbitrap Velos (Thermo Scientific) coupled with a NanoLC 1D HPLC (Eksigent). Samples were loaded onto a laboratory-made capillary column (9 cm long, 75 μm inner diameter), packed with Magic C18 AQ beads (3 μm, 100 Å, Microm) and eluted with a 5% to 40% v/v acetonitrile concentration gradient over 70 min, followed by 80% v/v acetonitrile for 10 min, at a flow rate of 0.25 μl min$^{-1}$. Peptide ions were detected in a full MS1 scan for mass-to-charge ratios between 300 and 2,000. MS2 scans were performed for the ten peptides with the highest MS signal (minimal signal strength 500 hits, isolation width mass-to-charge ratio 3 m/z, relative collision energy 35%). Peptide masses for which MS/MS spectra had been recorded were excluded from further MS/MS scans for 30 s.

## Peak area based protein quantification and statistical analysis

Quantitative analysis of MS/MS measurements was performed with Progenesis LCMS software (Nonlinear Dynamics). One run was selected as a reference, and for each run, 15 vectors were placed manually on prominent peaks before applying the automatic alignment and peak picking functions of Progenesis. Normalisation factors across all samples ranged between 0.7 and 1.4. The best eight spectra for each MS1 signal peak were exported to Mascot. Mascot search parameters were set as follows: *Arabidopsis* TAIR10 genome annotation, requirement for tryptic ends, one missed cleavage allowed, fixed modification: carbamidomethylation (cysteine), variable modification: oxidation (methionine), peptide mass tolerance = ± 10 ppm, MS/MS tolerance = ± 0.6 Da, allowed peptide charges of +2 and +3. Spectra were also searched against a decoy database of the *Arabidopsis* proteome and results were filtered to ensure a FDR below 1% on the protein level. Additionally, peptide identifications with a Mascot score below 25 were excluded. Mascot results were imported into Progenesis, quantitative peak area information extracted and the results exported for data plotting and statistical analysis. Mass spectrometry data used for quantification can be found on the EMBL proteomic repository PRoteomics IDEntifications (PRIDE; accession: PXD006848, https://doi.org/10.6019/pxd006848). This analysis was performed in R (version 3.2.3; R Core Team, 2015).

Statistical analysis to identify significantly changing proteins was performed in R (version 3.2.3; R Core Team, 2015) using log2-transformed relative abundance values. First, analysis of variance (ANOVA) was performed across photoperiods. The resulting *P*-values were corrected for multiple testing with the Benjamini–Hochberg method to control the global FDR. Next, significant changes between photoperiods were computed by pairwise comparison using the Tukey honest significant differences (TukeyHSD) post hoc test followed by correction with the Benjamini–Hochberg method. The results of this analysis for all proteins are presented in Table EV3.

Overrepresentation analysis of functional categories was performed using KEGG pathway annotations (Kanehisa *et al*, 2016) and gene ontology (GO) terms (Ashburner *et al*, 2000). *Arabidopsis* GO annotations were obtained from the Gene Ontology Consortium database (http://www.geneontology.org). Overrepresentation of GO terms and KEGG pathways was assessed using Fisher's exact test.

## Selection of arrhythmic transcripts

Reliably arrhythmic transcripts (i.e. transcripts with no detectable diurnal rhythm in transcript levels) were identified by applying a set of criteria based on available transcriptomic analysis from (Bläsing *et al*, 2005) and (Flis *et al*, 2016). Transcripts were identified as reliably arrhythmic if they were not in the set of diurnally rhythmic transcripts identified by ANOVA in (Bläsing *et al*, 2005), and if they had no significant difference between end-of-day and end-of-night expression in any of the 5 photoperiods examined by (Flis *et al*, 2016), as assessed by a two-tailed *t*-test at a $P = 0.05$ threshold (Bonferroni corrected for multiple testing across 5 photoperiods).

## Mathematical model of translational coincidence

We consider a simple model with different rates of translation in the light ($T_L$) and in the dark ($T_D$). For arbitrary mRNA dynamics given by $m(t)$, the daily rate of protein synthesis is then:

$$k_s = T_L \int_{t=0}^{t_{dusk}} m(t) + T_D \int_{t=t_{dusk}}^{24} m(t)$$

For slowly turning-over proteins, protein abundance reaches a steady state where synthesis is balanced by turnover ($k_d$) and dilution by growth ($\mu$):

$$P = \frac{k_s}{k_d + \mu}$$

Quantitative proteomics measures abundance relative to an internal standard. We assume that this internal standard can be

represented by the abundance of an "average" protein with no transcript rhythm and a turnover rate of $k_{d,reference}$, with its rate of synthesis given by:

$$k_{s,reference} = T_L t_{dusk} + T_D(24 - t_{dusk})$$

Its abundance is then given by:

$$P_{reference} = \frac{k_{s,reference}}{k_{d,reference} + \mu}$$

This represents the background changes in protein levels, against which changes in protein levels are normalised. Assuming that any given protein has a turnover similar to the background (i.e. $k_d = k_{d,reference}$), we obtain the normalised value analogous to that measured by quantitative proteomics:

$$P_{normalised} = \frac{P}{P_{reference}} = \frac{T_L \int_{t=0}^{t_{dusk}} m(t) + T_D \int_{t=t_{dusk}}^{24} m(t)}{T_L t_{dusk} + T_D(24 - t_{dusk})}$$

For a relative rate of protein synthesis in the light compared to the dark of $R$ ($= T_L/T_D$), this becomes:

$$P_{normalised} = \frac{R \int_{t=0}^{t_{dusk}} m(t) + \int_{t=t_{dusk}}^{24} m(t)}{(R - 1)t_{dusk} + 24}$$

This expresses the protein level at a given photoperiod ($t_{dusk}$) as a function of the measured transcript dynamics ($m(t)$) and the measured ratio of protein synthesis in the light compared to the dark ($R$). Based on $^{13}CO_2$ labelling data, this ratio was estimated to have a value of 1.4 (Pal *et al*, 2013).

We note that differences in the rate of protein turnover ($k_d$) between proteins will induce systematic deviations in this relationship. However, since there is no known systematic relationship between the timing of transcript expression and the rate of protein turnover, these systematic deviations are not expected to affect the relationship observed between transcript expression and photoperiod response.

Changes in protein level between two photoperiods are then compared relative to the mean abundance between those photoperiods:

$$\Delta P = \frac{\left(P_{normalised,2} - P_{normalised,1}\right)}{\left(\left(P_{normalised,1} + P_{normalised,2}\right)/2\right)}$$

This gives the model predictions used in Fig 6F.

### Inference of protein synthesis and degradation rates in Ostreococcus and Cyanothece

Synthesis and degradation rates for Ostreococcus and Cyanothece proteins were calculated from the proteomics time series datasets of (Martin *et al*, 2012) and (Aryal *et al*, 2011), respectively. These datasets characterised the dynamics of partial stable isotope incorporation with $^{15}N$ (Ostreococcus) and heavy leucine (Cyanothece) during several days of growth in light/dark cycles. For each species, we inferred a labelling efficiency from the maximum labelled fraction achieved of any protein, which was equal to 0.93 for Ostreococcus and

to 0.8 for Cyanothece. To infer degradation rates, we fitted a simple kinetic model assuming (i) constant labelling efficiency over time; (ii) different proteins are labelled at the same efficiency; (iii) heavy and light fractions are turned over at equal rates. To infer relative rates of synthesis in the light and dark, we took the average ratio of labelling rates between time-points spanning the light and dark periods.

**Data and software availability**

The datasets and computer code produced in this study are available in the following databases:

- Quantitative proteomics data: PRIDE; accession: PXD006848, https://doi.org/10.6019/pxd006848.
- Code and data for simulating the translational coincidence model: FAIRDOM HUB; ID: 163; URL: https://fairdomhub.org/investigations/163. DOI: https://doi.org/10.15490/fairdomhub.1.investigation.163.2

**Expanded View** for this article is available online.

### Acknowledgements

Research was supported by the European Union (FP7 collaborative project TiMet, contract no. 245143), by BBSRC award BB/D019621/1 to AJM, by ETH Zurich (Switzerland) and by the Max Planck Society (Germany).

### Author contributions

Designed research (DDS, AG, MS, AJM, WG); performed proteomic measurements (AG, KB), analysed data and prepared figures (AG, DDS); conceived and performed modelling (DDS); wrote the paper (DDS and AG with input from all authors).

### Conflict of interest

The authors declare that they have no conflict of interest.

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
