## [Review Process File · Molecular Systems Biology]

Photoperiodic control of the Arabidopsis proteome reveals a translational coincidence mechanism

Daniel D Seaton, Alexander Graf, Katja Baerenfaller, Mark Stitt, Andrew J. Millar and Wilhelm Gruissem

Review timeline:

Submission date:	29 August 2017
Editorial Decision:	15 October 2017
Revision received:	5 December 2017
Editorial Decision:	11 January 2018
Revision received:	22 January 2018
Accepted:	30 January 2018

Editor: Thomas Lemberger

Transaction Report:

1st Editorial Decision

15 October 2017

Thank you again for submitting your work to Molecular Systems Biology. We have now heard back from the two referees who agreed to evaluate your manuscript. As you will see from the reports below, the referees find the topic of your study of potential interest. They raise, however, several points which should be convincingly addressed in a revision of the manuscript. The recommendations provided by the reviewers are very clear in this regard.

We would also kindly ask you to include after the Materials & Methods section a formal "Data and Software Availability" section that follows the example below:

#Data and software availability

The datasets and computer code produced in this study are available in the following databases:

- RNA-Seq data: Gene Expression Omnibus GSE46843
[<https://www.ncbi.nlm.nih.gov/geo/query/acc.cgi?acc=GSE46843>]
- Chip-Seq data: Gene Expression Omnibus GSE46748
[<https://www.ncbi.nlm.nih.gov/geo/query/acc.cgi?acc=GSE46748>]
- Protein interaction AP-MS data: PRIDE PXD000208
[<http://www.ebi.ac.uk/pride/archive/projects/PXD000208>]
- Imaging dataset: Image Data Resource doi:10.17867/10000101
[<http://doi.org/10.17867/10000101>]
- Modeling computer scripts: GitHub
[<https://github.com/SysBioChalmers/GECKO/releases/tag/v1.0>]
- [data type]: [full name of the resource] [accession number/identifier] ([doi or URL or

identifiers.org/DATABASE:ACCESSION])

If you feel you can satisfactorily deal with these points and those listed by the referees, you may wish to submit a revised version of your manuscript. Please attach a covering letter giving details of the way in which you have handled each of the points raised by the referees. A revised manuscript will be once again subject to review and you probably understand that we can give you no guarantee at this stage that the eventual outcome will be favorable.

 REVIEWER REPORTS

Reviewer #1:

This article addresses a novel and important question, how protein synthesis and gene expression in general are affected by seasonality (day-length). The article presents a proteomic dataset, collected from *Arabidopsis* vegetative rosettes under four different day-length regimes (6h-18h), which corresponds approximately to plants growing in winter (temperature permitting) to summer at a northern latitude.

On the basis of the new proteomic data, existing transcriptome data from the same group(s), and other data mined from the literature, the authors present an intriguing hypothesis: That day-length influences protein synthesis by a general mechanism where light in the evening (but not in morning) stimulates the protein synthesis from a specific subset of transcripts, namely those that happen to have a clock-controlled peak in the evening (not in the morning). This hypothesis, which they term "translational coincidence", is an extension of the much more specific and well established 'external coincidence' model that describes our understanding of the photoperiodic (seasonal) control of flowering.

Overall, it is a stimulating paper on a topic appropriate for MSB. I commend the authors for their astute observation, the creative conjecture from their data, and the motivation for a mathematical treatment.

General comments:

The data appear to be of good quality, although documentation needs to be improved (see below). After presenting several figures of basal descriptive analysis, which is necessary to convince us of the quality of the data, the core figure is Fig. 6C. It shows how mRNAs with a peak transcript level late in the day (ZT12) tend to code for proteins whose protein level is enhanced by day length, whereas mRNAs that peak in the morning do not show this day-length dependence. The data are not being overinterpreted (line 305 and line 372 "our data are consistent with ...") although the authors fairly explore the limits of how far they can push. The authors then argue elaborately that this phenomenon can be explained by the well-supported fact that translational efficiency is higher during the light period than in the dark. This is a reasonable explanation, and in fact one might even go as far as saying that - given preexisting data - one should be able to infer deductively that protein synthesis should follow the rule that the authors have discovered here within their new data. My main concern rests on the assumption, which is implicitly made in the manuscript, that the pattern observed should be explained solely by light-stimulated translation, and that a role for light stimulated transcription can be discounted. In fact, the result presented could potentially be explained by a 'transcriptional coincidence' mechanism, where light over the course of a long day (18h of light) simply drives higher mRNA levels than light on a short day (8h light). The authors do not present the crucial data to thoroughly and directly discount this idea, which would be to measure mRNA abundance over the course of a 24h cycle in all four photoperiods. Instead, the authors (being fully aware of the constraints) are engaging in extensive data mining of existing data to show that (i) the basic phenomenon they describe can be seen in prior studies (EV4 and EV5), (ii) that the foundation for the translational coincidence model is also present in cyanobacteria/algae (Fig. 7), and (iii) that existing transcript data show no evidence for a transcriptional coincidence effect (EV6, EV7, EV8, noting that absence of evidence does not equal evidence of absence).

Major comments

Line 134. I assume you harvested the entire rosettes without the roots?

Line 167: For functional analysis of ranked lists like yours, Gene Set Enrichment Analysis can be a very sensitive tool (GSEA). If you have not tried it, it may be worth the effort.

Figure EV3. Here it seems to me that we see evidence that transcript levels correlate with protein levels, as one might expect. In panel A, it is unclear how the fold-changes are defined. Are they between LPP 18h and SPP 8h? Always, for both protein and transcript? Why do you think the slope of the regression is <1 (it is about 1 in panel B).

Line 272: It is important to consider transcription of clock-regulated genes also depends on the light environment (as acknowledged in Fig. 5A). There are many genes in Arabidopsis whose mRNA cycle differs between a purely clock-driven cycle (as in LL) and a clock +light-driven cycle (as in LD). The mRNAs picked out for Figure EV6 are all clock-mRNAs and are somewhat unusual because, aside from CCA1, to my knowledge they have fairly limited light responsiveness. As I elaborated earlier in my synopsis, light-responses may be day-length dependent. In addition, translation of light-regulated mRNAs may also depend on the clock.

Line 298.

a) The Bläsing dataset comes from just one photoperiod, not 4.

b) The Flis dataset does not have full 24 cycles, just EOD and EON. Also, this manuscript did not clearly state how photoperiod affects absolute transcript levels in the Flis dataset.

Ideally, we would just want to look at genes that have exactly the same transcript dynamics in all four photoperiods. I'm afraid whether perhaps there are very few genes that would pass this filter? Is that the reason why it is difficult to conclusively rule out a "transcriptional coincidence" effect?

Line 720. I did not quite follow the reasoning here, but there seems to be some conjecture involved in interpreting the algal proteomics data from the literature. Since the algae were not analyzed experimentally for their photoperiod-dependent proteomes, I would be more comfortable if this section of the manuscript was omitted.

Tables: All the data tables require more extensive legends that explain exactly what is shown, spell out abbreviations, etc.

Minor comments

Abstract: The last abstract does not follow from what was said earlier.

Line 90-91. Fix syntax. Is a word missing?

Line 92. Clarify what exactly you mean by 'These'.

Line 96. As you are searching for adaptive value of circadian clocks, you seem to discount the benefit of being able to anticipate dawn independent of the length of the night.

Line 311: Instead of 'lower abundance' do you mean 'repression by a longer photoperiod'?

Line 426. The sweeping statement may be considered arguable by some in the field.

Line 463. omit 'during'.

Line 481. Omit sentence 'These integrate...' because this is vague/ not well supported.

Line 568. Omit 'it'.

Line 576. Do you mean 'when new synthesis of the protein is most needed'?

Reviewer #2:

Summary

In this manuscript, Seaton et al. propose the idea of "Translational Coincidence" a coordination between transcriptional control by the circadian clock and the general increase in translation rates in

long days in photosynthetic organisms. They hypothesize that this translational coincidence would selectively alter the protein abundance of select transcripts in a photoperiod dependent manner. The authors predicted that the interaction between transcripts that are rhythmically expressed and this enhanced translation rate in long day photoperiods could control the production levels of specific proteins in a seasonally adjusted manner. Transcripts with expression peaking at the end of the day will have lower protein abundance levels in shorter photoperiods and will increase in longer photoperiods. In contrast, the prediction is that translation of genes with a peak of expression around dawn will be independent of photoperiod and will therefore maintain consistent protein levels across photoperiods. To assay this, in *Arabidopsis*, they examined the abundance of proteins grown in different photoperiods at the end of the day and compared the changes in protein levels to the timing of the transcriptional peak of expression. They observed that for transcripts with a peak phase of expression in the evening there is an increase in protein abundance with increasing photoperiod. Based on this information, a model of protein abundance that incorporates the specific gene's predicted sensitivity to photoperiod was developed and tested. This model allowed the prediction of each gene's protein abundance across the photoperiods (tdusk), which the authors evaluated against their experimental data. The model generally showed a positive trend in the ability to predict the change in protein level across photoperiods. Finally, the authors speculate on the prevalence of this mechanism across photosynthetic organisms by examining the presence of conserved components they hypothesize are necessary in *O. tauri* and *Synechococcus*.

The proposed concept is novel and although many aspects will need to be tested, the presented data provides clear support to the idea. This is a significant conceptual advance, which proposes a whole new role for the transcriptional regulation of genes by the circadian clock. While the presented research supports this concept in *Arabidopsis*, the authors speculate on its extension into other photosynthetic organisms, providing evidence for the required components. In the discussion they further speculate on a broader application across diverse organisms. Therefore, this concept and the evidence presented here in support of it will likely be of interest to a wide audience.

Major points

1. Is the increase in protein synthesis associated with photoperiod truly "without consequence" for transcripts peaking at dawn / just before dawn? The authors identify 1024 proteins with lower abundance in long photoperiods, more than increase in abundance. Figure 6B indicates that these are (slightly?) enriched for an increase in genes that are phased at dawn. It appears that they may be decreasing across photoperiods (eg. EV4A). In figure EV5A it appears that in an independent data set their decrease is as relatively strong as the increase observed in transcripts with a peak between 8-16. It may just be due to the presentation of the data as normalized to the total protein level, therefore if more of the new proteins made in longer photoperiods are from evening phased transcripts, by necessity, the AM phased- transcripts would decrease in percent abundance even if their abundance remained the same. Yet it is not clear if that is the case - in figure EV4 it appears the total decrease is greater than the increase in evening phased genes (could be due to differences in the number of genes OR really a difference in abundance level). Do transcripts with a dawn phase of expression remain consistent in abundance levels, how about non-cycling transcripts? What information is gained from the predicted model for dawn-phased transcripts / non-cycling transcripts? This information would strengthen the predictions for "translational coincidence" if supportive, and if not supportive might imply more complicated regulation, either way it would be useful to understand if the apparent decrease in relative abundance is due to the necessity of normalizing to the total protein or if this is an additional regulatory layer.

2. It is interesting that the GO enrichment of proteins that change in abundance across photoperiods are restricted to such narrow fold-change levels. Suggesting that the stoichiometry of these categories is tightly regulated at the protein level. From Figure 2B there are a few categories that are broadly distributed across multiple fold-change categories. Yet examination of the table in EV6 (see note 2 below about table labeling), shows some categories with enrichment across more fold change levels. A more detailed description of the table of EV6 or Figure 2B could resolve the following questions: Why do the FC levels in Figure 2 B & C stop at 1.5, yet, EV6 has larger differences, are there no significant ones above 1.5? Is EV6 all changes, not just significant? Why is there a difference in the breadth of enrichment for categories across fold change levels between table EV6 and Figure 2B: For example, GO:0009684 appears across multiple f.c. levels in EV6, yet the most enriched category in figure 2B shows up in 3 levels. Again, perhaps this is due to all changes vs.

"significantly changing" proteins, but this is not clear from the text, legends, or materials and methods.

The goal and outcomes of the reduction in the examined proteins (starting at Line 343) to remove the potentially confounding effects of transcriptional regulation is not clear. The logic for doing this is well outlined, and the approach is clearly justified, however, the outcome of this step and the information gained from this is not obvious to this reviewer. The retention of the observed pattern supporting translational coincidence is clear, but I would expect that this would improve the observed increase in evening phased sensitivity to photoperiods if you remove the ones with confounding effects. It would be helpful to clarify the point of this reduction and what conclusions can be drawn from comparing the reduced categories, not just pull out a few examples. Was the model re-trained or re-run on this reduced sample? Were the results different? Or perhaps what would figure 6B look like for just these genes?

Minor points

1. It is this reviewer's opinion that figure EV1 and especially figureEV4 are very informative and supportive of the conclusions presented and should be included in the main text if possible. Figure 1 C&D, and Figure 2 B&C (as there is no association with what the GO terms are) are less informative and could be moved to supplemental if needed for space constraints.

2. There is some confusion with labeling of the supplemental files. The labels within the excel sheets themselves do not match up with the link they are saved under.

3. Line 88: Is it possible that CBF1 should be CDF1, since only the latter has been used to build quantitative models (although both show photoperiod sensitivity)?

4. Line 232: A GO term overrepresentation analysis using photoperiod responsive proteins with lower abundance in long photoperiods revealed that most of the 39 significantly enriched GO categories are related to transcription, translation and cell cycle (Table EV2).

>Table EV2 (KEGG pathways), does not relate to this statement it is not clear if any of the supplementary tables show the indicated results. (see point 2 about confusion in labeling, but I could not find any table that fit the point in the text).

5. Line 250: How many transcript-protein pairs showed significant changes in both transcript and protein levels with the given cutoffs? Appears to be many from graph.

6. The pdf versions of the supplemental data provided were not formatted correctly and only contained information from one sheet of the original .xlsx files.

To the Editor,

We thank the reviewers for their thoughtful and constructive assessment of our manuscript.

We have made a number of changes to address the comments made by the reviewers. The key additions are new analyses to account for potential transcriptional regulation that reviewers were concerned may confound our interpretation of our proteomics dataset. Here, we used a published photoperiod microarray time series dataset from (Michael et al, 2008b) to further demonstrate that confounding transcriptional effects are unlikely to explain the observed relationship between circadian phase and protein abundance response to photoperiod (Fig EV5D,E). We also provide additional plots illustrating that our mathematical model fits well the filtered subsets of genes without any potentially confounding transcriptional regulation (EV5F).

Our detailed response to the reviewers is provided below. Our responses are in italics, with corresponding new text in the manuscript in quotation marks.

Reviewer #1:

This article addresses a novel and important question, how protein synthesis and gene expression in general are affected by seasonality (day-length). The article presents a proteomic dataset, collected from *Arabidopsis* vegetative rosettes under four different day-length regimes (6h-18h), which corresponds approximately to plants growing in winter (temperature permitting) to summer at a northern latitude.

On the basis of the new proteomic data, existing transcriptome data from the same group(s), and other data mined from the literature, the authors present an intriguing hypothesis: That day-length influences protein synthesis by a general mechanism where light in the evening (but not in morning) stimulates the protein synthesis from a specific subset of transcripts, namely those that happen to have a clock-controlled peak in the evening (not in the morning). This hypothesis, which they term "translational coincidence", is an extension of the much more specific and well established 'external coincidence' model that describes our understanding of the photoperiodic (seasonal) control of flowering.

Overall, it is a stimulating paper on a topic appropriate for MSB. I commend the authors for their astute observation, the creative conjecture from their data, and the motivation for a mathematical treatment.

General comments:

The data appear to be of good quality, although documentation needs to be improved (see below). After presenting several figures of basal descriptive analysis, which is necessary to convince us of the quality of the data, the core figure is Fig. 6C. It shows how mRNAs with a peak transcript level late in the day (ZT12) tend to code for proteins whose protein level is enhanced by day length, whereas mRNAs that peak in the morning do not show this day-length dependence. The data are not being overinterpreted (line 305 and line 372 "our data are consistent with ...") although the

authors fairly explore the limits of how far they can push. The authors then argue elaborately that this phenomenon can be explained by the well-supported fact that translational efficiency is higher during the light period than in the dark. This is a reasonable explanation, and in fact one might even go as far as saying that - given preexisting data - one should be able to infer deductively that protein synthesis should follow the rule that the authors have discovered here within their new data. My main concern rests on the assumption, which is implicitly made in the manuscript, that the pattern observed should be explained solely by light-stimulated translation, and that a role for light stimulated transcription can be discounted. In fact, the result presented could potentially be explained by a 'transcriptional coincidence' mechanism, where light over the course of a long day (18h of light) simply drives higher mRNA levels than light on a short day (8h light). The authors do not present the crucial data to thoroughly and directly discount this idea, which would be to measure mRNA abundance over the course of a 24h cycle in all four photoperiods. Instead, the authors (being fully aware of the constraints) are engaging in extensive data mining of existing data to show that (i) the basic phenomenon they describe can be seen in prior studies (EV4 and EV5), (ii) that the foundation for the translational coincidence model is also present in cyanobacteria/algae (Fig. 7), and (iii) that existing transcript data show no evidence for a transcriptional coincidence effect (EV6, EV7, EV8, noting that absence of evidence does not equal evidence of absence).

We would like to clarify an important point regarding 'transcriptional coincidence', and our analysis to take this into account. There are some widely known examples of this effect (e.g. the classic FT response, which is dependent on light-stabilisation of CO, an upstream transcriptional activator). The purpose of our analysis of orthogonal transcriptional datasets is to delimit a subset of transcripts which we are confident are not subject to this confounding effect, and show that the proteomics data show the same relationship between phase of transcript expression and change in protein abundance.

As pointed out by the reviewer, the ideal method for controlling for this would be to have RNA time series datasets from plants grown in the same conditions. We now include an additional analysis of a published dataset: the microarray time series from seedlings (Ler accession) grown in 8h and 16h photoperiods (Michael et al, 2008). Applying a further filter according to the fold-change averaged across these time series identifies a subset of proteins for which there is no detectable relationship between transcript phase and transcript photoperiod change (New Fig EV5E), while the relationship between transcript phase and protein is still very clear (New Fig EV5C,D). This is now described in detail in the text (line 373):

“Third, we compared mean expression levels in short (8h) and long (16h) photoperiods, using the transcriptome timeseries dataset of (Michael et al., 2008b) (data from 2-week old seedlings, Landsberg (Ler) accession). We identified transcripts with photoperiod-dependent changes in mean expression levels in this dataset and filtering out transcripts with a greater than 0.3-fold change in expression. This filter reduced the number of proteins considered from 547 to 378. The pattern of photoperiod response remained the same after this filtering (Fig EV5C). Furthermore, this pattern remained after combining all three filters, reducing the total number of proteins considered further to 104 (Fig EV5D). The transcripts in this set displayed no discernible relationship between phase of expression and change in abundance in the dataset of (Michael et al., 2008b) (Fig EV5E). The translational coincidence model maintained a good fit to the data for this subset (Pearson's $R = 0.5$, $p < 10^{-5}$) (Fig EV5F).”

Major comments

Line 134. I assume you harvested the entire rosettes without the roots?

Yes. We have clarified this ("Full rosettes were harvested" (line 132)).

Line 167: For functional analysis of ranked lists like yours, Gene Set Enrichment Analysis can be a very sensitive tool (GSEA). If you have not tried it, it may be worth the effort.

We performed GSEA analysis with our ranked list. This provided very similar results (e.g. GO terms for glycoside biosynthesis, ribosome biogenesis, lipid metabolism). Given the similarity, and since we have the KEGG pathway analysis as well as multiple GO enrichment analyses (including in the fold-change windows), we decided not to include these results.

Figure EV3. Here it seems to me that we see evidence that transcript levels correlate with protein levels, as one might expect. In panel A, it is unclear how the fold-changes are defined. Are they between LPP 18h and SPP 8h? Always, for both protein and transcript? Why do you think the slope of the regression is <1 (it is about 1 in panel B).

We have included the slopes of each line-of-best-fit in the plots. The fold change has been defined in the legend (line 852) - "Fold changes are defined as the maximal fold change measured across photoperiods (Table EV3)". The regression slopes are relatively close to 1 - ranging from 1.14 to 0.74. Deviations from 1 are not straightforward to interpret, since they result from a combination of biological factors (e.g. buffering of protein changes by post-translational mechanisms, such as protein-protein interactions) and technical factors (e.g. compression of fold-changes detected by proteomics measurements), and we have avoided making any assertions in this direction.

Line 272: It is important to consider transcription of clock-regulated genes also depends on the light environment (as acknowledged in Fig. 5A). There are many genes in Arabidopsis whose mRNA cycle differs between a purely clock-driven cycle (as in LL) and a clock +light-driven cycle (as in LD). The mRNAs picked out for Figure EV6 are all clock-mRNAs and are somewhat unusual because, aside from CCA1, to my knowledge they have fairly limited light responsiveness. As I elaborated earlier in my synopsis, light-responses may be day-length dependent. In addition, translation of light-regulated mRNAs may also depend on the clock.

We have clarified in the text that the purpose of EV6 is just to check the criterion we apply for identifying transcripts that are predominantly regulated by the circadian clock and not by other factors (line 361):

"As confirmation that this criterion identifies transcripts under strong circadian regulation, we note that, as expected, the core circadian clock genes CCA1, PRR7, ELF3, and TOC1 all pass this test (Fig EV4)."

We have further dealt with potential confounding effect by light-driven changes in transcriptional regulation by adding an additional filter based on the microarray dataset of (Michael et al, 2008b), as described above, and Fig EV5.

Line 298.

a) The Bläsing dataset comes from just one photoperiod, not 4.

b) The Flis dataset does not have full 24 cycles, just EOD and EON. Also, this manuscript did not clearly state how photoperiod affects absolute transcript levels in the Flis dataset.

Ideally, we would just want to look at genes that have exactly the same transcript dynamics in all four photoperiods. I'm afraid whether perhaps there are very few genes that would pass this filter? Is that the reason why it is difficult to conclusively rule out a "transcriptional coincidence" effect?

We have reworded Line 298 to clarify that the Bläsing dataset is from only one (12h) photoperiod (now line 303). We also removed the term "pseudo-time series" from the Flis dataset (now line 357).

Line 720. I did not quite follow the reasoning here, but there seems to be some conjecture involved in interpreting the algal proteomics data from the literature. Since the algae were not analyzed experimentally for their photoperiod-dependent proteomes, I would be more comfortable if this section of the manuscript was omitted.

We have added a clarification of the limitations of this analysis at the introduction to this section in the Results (line 404):

"These datasets do not characterise the photoperiodic responses of the proteomes and transcriptomes of these species. Nevertheless, they allow assessment of whether light-induced protein synthesis and circadian regulation of gene expression are observed in these cases."

We note that Reviewer #2 highlighted this section as important for establishing the potential relevance of our results to a broad range of organisms, and prefer to keep this section of the manuscript for this reason.

Tables: All the data tables require more extensive legends that explain exactly what is shown, spell out abbreviations, etc.

We have significantly expanded the legends, and modified column headings, to make these clearer.

Minor comments

Abstract: The last abstract does not follow from what was said earlier.

While not immediately following from the previous statements, we felt that this was an important interpretation of our results, and understandable for readers in this area.

Line 90-91. Fix syntax. Is a word missing?

Line 92. Clarify what exactly you mean by 'These'.

Line 96. As you are searching for adaptive value of circadian clocks, you seem to discount the benefit of being able to anticipate dawn independent of the length of the night.

We have reworded the section from lines 90-96 to clarify how the specific examples (CO, FKF1, etc) work, and how these illustrate a potential role for circadian regulation in photoperiod responses,

without implying discounting of the alternatives (such as anticipation of dawn). Specifically, we note (line 98):

“Their canonical phenotypes, especially seasonal reproduction, are the most important, known effects of plant circadian regulation. This suggests a potential role for photoperiod responses as a driving force for the evolution of pervasive circadian regulation across the genome (Millar, 2016).”

Instead of:

“Their canonical phenotypes, especially seasonal reproduction, are the most important, known effects of plant circadian regulation. But it seems unlikely that these few traits account for the evolution of pervasive, circadian regulation across the genome (Millar, 2016).”

Line 311: Instead of 'lower abundance' do you mean 'repression by a longer photoperiod'?

We have left this as is, for consistency throughout the paper - we have generally used the terms 'higher' and 'lower' abundance rather than “activated”/“repressed”.

Line 426. The sweeping statement may be considered arguable by some in the field.

We have re-worded the start of the Discussion to mitigate this (line 457), while still highlighting the value of our proteomics dataset for studying systems-level changes:

“Here, we quantified the response of the Arabidopsis proteome to changing photoperiods, providing a broad view of photoperiodic responses across many pathways and processes.”

Line 463. omit 'during'.

Line 481. Omit sentence 'These integrate...' because this is vague/ not well supported.

Line 568. Omit 'it'.

We have made these suggested changes.

Line 576. Do you mean 'when new synthesis of the protein is most needed'?

We have removed this sentence (line 594), to focus on the case where protein rhythms are not observed despite transcript rhythms being present (i.e. the pervasive case).

Reviewer #2:

Summary

In this manuscript, Seaton et al. propose the idea of "Translational Coincidence" a coordination between transcriptional control by the circadian clock and the general increase in translation rates in long days in photosynthetic organisms. They hypothesize that this translational coincidence would selectively alter the protein abundance of select transcripts in a photoperiod dependent manner. The

authors predicted that the interaction between transcripts that are rhythmically expressed and this enhanced translation rate in long day photoperiods could control the production levels of specific proteins in a seasonally adjusted manner. Transcripts with expression peaking at the end of the day will have lower protein abundance levels in shorter photoperiods and will increase in longer photoperiods. In contrast, the prediction is that translation of genes with a peak of expression around dawn will be independent of photoperiod and will therefore maintain consistent protein levels across photoperiods. To assay this, in *Arabidopsis*, they examined the abundance of proteins grown in different photoperiods at the end of the day and compared the changes in protein levels to the timing of the transcriptional peak of expression. They observed that for transcripts with a peak phase of expression in the evening there is an increase in protein abundance with increasing photoperiod. Based on this information, a model of protein abundance that incorporates the specific gene's predicted sensitivity to photoperiod was developed and tested. This model allowed the prediction of each gene's protein abundance across the photoperiods (tdusk), which the authors evaluated against their experimental data. The model generally showed a positive trend in the ability to predict the change in protein level across photoperiods. Finally, the authors speculate on the prevalence of this mechanism across photosynthetic organisms by examining the presence of conserved components they hypothesize are necessary in *O. tauri* and *Synechococcus*.

The proposed concept is novel and although many aspects will need to be tested, the presented data provides clear support to the idea. This is a significant conceptual advance, which proposes a whole new role for the transcriptional regulation of genes by the circadian clock. While the presented research supports this concept in *Arabidopsis*, the authors speculate on its extension into other photosynthetic organisms, providing evidence for the required components. In the discussion they further speculate on a broader application across diverse organisms. Therefore, this concept and the evidence presented here in support of it will likely be of interest to a wide audience.

Major points

1. Is the increase in protein synthesis associated with photoperiod truly "without consequence" for transcripts peaking at dawn / just before dawn? The authors identify 1024 proteins with lower abundance in long photoperiods, more than increase in abundance. Figure 6B indicates that these are (slightly?) enriched for an increase in genes that are phased at dawn. It appears that they may be decreasing across photoperiods (eg. EV4A). In figure EV5A it appears that in an independent data set their decrease is as relatively strong as the increase observed in transcripts with a peak between 8-16. It may just be due to the presentation of the data as normalized to the total protein level, therefore if more of the new proteins made in longer photoperiods are from evening phased transcripts, by necessity, the AM phased- transcripts would decrease in percent abundance even if their abundance remained the same. Yet it is not clear if that is the case - in figure EV4 it appears the total decrease is greater than the increase in evening phased genes (could be due to differences in the number of genes OR really a difference in abundance level). Do transcripts with a dawn phase of expression remain consistent in abundance levels, how about non-cycling transcripts? What information is gained from the predicted model for dawn-phased transcripts / non-cycling transcripts? This information would strengthen the predictions for "translational coincidence" if supportive, and if not supportive might imply more complicated regulation, either way it would be useful to understand if the apparent decrease in relative abundance is due to the necessity of normalizing to the total protein or if this is an additional regulatory layer.

We have clarified in the text that translational coincidence is expected to affect the abundance (but not the rate of synthesis) of proteins with dawn-phased transcripts, and, indeed, the data show this effect clearly (Fig 6). line 311:

“In particular, we note that the lower abundance of proteins with dawn-phased transcripts in long photoperiods follows from the increased rate of dilution of all proteins by increased growth rate in long photoperiods (see below and Models and methods: Mathematical model of translational coincidence for details).”

Furthermore, the translational coincidence model correctly predicts this regulation by accounting for dilution by increased growth rate. We have clarified in the text that the model predicts protein abundance by calculating protein synthesis, and then accounting for dilution by growth (line 327):

“We next simulated translational coincidence in a quantitative model, using both transcript dynamics (Bläsing et al., 2005) and changes in bulk protein synthesis rates measured by ¹³CO₂ labelling (Pal et al., 2013) to predict changes in protein synthesis, and consequently abundance, across photoperiods (Fig 6E). This model accounts for changes in protein synthesis depending on light and mRNA abundance. Changes in protein synthesis then result in changes in abundance after accounting for changes in the rate of bulk protein synthesis with photoperiod (i.e. accounting for changes in growth with photoperiod) (see Models and methods: Mathematical model of translational coincidence for details). Similar approaches have previously been applied to model isotope labelling of protein in growing tissues (Ishihara et al., 2015). Protein abundance in a specified photoperiod is then given by:”

2. It is interesting that the GO enrichment of proteins that change in abundance across photoperiods are restricted to such narrow fold-change levels. Suggesting that the stoichiometry of these categories is tightly regulated at the protein level. From Figure 2B there are a few categories that are broadly distributed across multiple fold-change categories. Yet examination of the table in EV6 (see note 2 below about table labeling), shows some categories with enrichment across more fold change levels. A more detailed description of the table of EV6 or Figure 2B could resolve the following questions: Why do the FC levels in Figure 2 B & C stop at 1.5, yet, EV6 has larger differences, are there no significant ones above 1.5? Is EV6 all changes, not just significant? Why is there a difference in the breadth of enrichment for categories across fold change levels between table EV6 and Figure 2B: For example, GO:0009684 appears across multiple f.c. levels in EV6, yet the most enriched category in figure 2B shows up in 3 levels. Again, perhaps this is due to all changes vs. "significantly changing" proteins, but this is not clear from the text, legends, or materials and methods.

We have improved the EV Table legends to clarify the information contained in each table, and corrected mistaken references to the EV Tables. The table referred to is now Table EV5 - GO enrichment by FC window. For Figure 2 B & C, we have visualised only the smaller fold changes, to highlight the (to our mind) surprising observation that GO enrichments are localised to specific fold change windows. This is now explained in the legend (line 852). Table EV5 shows the p-values from enrichment tests (as now specified in the legend - line 1046), and while there are a few significant

enrichments at higher fold change windows, these are very few and sparse, and including these in Figure 2 B & C added little.

3. The goal and outcomes of the reduction in the examined proteins (starting at Line 343) to remove the potentially confounding effects of transcriptional regulation is not clear. The logic for doing this is well outlined, and the approach is clearly justified, however, the outcome of this step and the information gained from this is not obvious to this reviewer. The retention of the observed pattern supporting translational coincidence is clear, but I would expect that this would improve the observed increase in evening phased sensitivity to photoperiods if you remove the ones with confounding effects. It would be helpful to clarify the point of this reduction and what conclusions can be drawn from comparing the reduced categories, not just pull out a few examples. Was the model re-trained or re-run on this reduced sample? Were the results different? Or perhaps what would figure 6B look like for just these genes?

We have evaluated the model predictions on a reduced set of genes, and found similar results, plotted in (EV5F). This is also similar for all other subsets plotted in Fig EV5A,B,C (data not shown). There is indeed an improved fit when taking these subsets of genes, but it is modest (from $R=0.4$ to $R=0.5$). We note that the model has no adjustable parameters to train; it only uses measured parameters. These are the measured (relative, bulk) translation rates, and measured transcript abundance. In addition, we applied an additional filter based on the transcript dataset of Michael et al, 2008b (see response to Reviewer #1).

Minor points

1. It is this reviewer's opinion that figure EV1 and especially figure EV4 are very informative and supportive of the conclusions presented and should be included in the main text if possible. Figure 1 C&D, and Figure 2 B&C (as there is no association with what the GO terms are) are less informative and could be moved to supplemental if needed for space constraints.

We agree that these plots are informative, and have included them in the main figure panels, while rearranging other panels to make space.

2. There is some confusion with labeling of the supplemental files. The labels within the excel sheets themselves do not match up with the link they are saved under.

We have corrected the referencing and labelling of the supplementary tables, and apologise for any confusion caused.

3. Line 88: Is it possible that CBF1 should be CDF1, since only the latter has been used to build quantitative models (although both show photoperiod sensitivity)?

This is correct - we have changed this.

4. Line 232: A GO term overrepresentation analysis using photoperiod responsive proteins with lower abundance in long photoperiods revealed that most of the 39 significantly enriched GO categories are related to transcription, translation and cell cycle (Table EV2).

>Table EV2 (KEGG pathways), does not relate to this statement it is not clear if any of the supplementary tables show the indicated results. (see point 2 about confusion in labeling, but I could not find any table that fit the point in the text).

The reviewer is correct - a table was missing. We apologise for the confusion. We have now included a table containing the complete GO results, and have corrected the references to the various different GO/KEGG enrichment results tables.

5. Line 250: How many transcript-protein pairs showed significant changes in both transcript and protein levels with the given cutoffs? Appears to be many from graph.

We have modified the legend to clarify that the number stated in the title of each panel is the number of significant changes shown. We now also mention the key numbers explicitly in the text (line 255):

“Using these conditions, 421 and 390 transcript:protein pairs were selected at ED and EN, respectively.”

6. The pdf versions of the supplemental data provided were not formatted correctly and only contained information from one sheet of the original .xlsx files.

We were not able to figure out how to correct this in the conversion process, and assume that only the .xlsx files will be provided with the final manuscript.

2nd Editorial Decision

11 January 2017

Thank you again for submitting your work to Molecular Systems Biology. We are now satisfied with the modifications made. I am pleased to inform you that we will be able to publish your paper in Molecular Systems Biology pending the following minor amendments:

Main files

- Please replace the manuscript PDF file with a word file.
- Please upload high quality individual figures files for all main figures and EV figures.
- Please remove the figure legends from the figures and EV figures. The legends should only appear in the manuscript file.
- Please move the EV table legends from the manuscript file to individual tabs in the corresponding table excel files.
- the synopsis image provide is very pixelated and too low resolution (157x121 at 72dpi). We would be grateful if you could provide an image with the absolute dimensions width=211 x height=157 pixels.

Supplemental text

- Please rename the file from Supplemental text to Appendix and add a Table of Contents on the first page.

Callouts

- Please add explicit callouts to Appendix fig. S9 and S10 in the main text or at least in the Appendix. These figures do not seem to be referenced anywhere.

2nd Revision - authors' response

22 January 2018

Many thanks for your consideration of our manuscript.

The following requested changes to our paper have now been addressed:

Main files

- Please replace the manuscript PDF file with a word file.
- Please upload high quality individual figures files for all main figures and EV figures.
- Please remove the figure legends from the figures and EV figures. The legends should only appear in the manuscript file.
- Please move the EV table legends from the manuscript file to individual tabs in the corresponding table excel files.
- the synopsis image provide is very pixelated and too low resolution (157x121 at 72dpi). We would be grateful if you could provide an image with the absolute dimensions width=211 x height=157 pixels.

Supplemental text

- Please rename the file from Supplemental text to Appendix and add a Table of Contents on the first page.

Callouts

- Please add explicit callouts to Appendix fig. S9 and S10 in the main text or at least in the Appendix. These figures do not seem to be referenced anywhere.

Two notes:

- For the EV table legends, we understood this to mean having a separate 'legend' sheet for each excel file, and have added these to every EV Table.

- For the synopsis image, we now provide a 211x157 pixel vector format (pdf) image file rather than a bitmap format, as all bitmap formats (png, jpeg, etc) we were able to produce of this size were pixelated.

Corresponding Author Names: Andrew J. Millar and Wilhelm Grussem

Manuscript Number: MSB-17-7962R